# Metallic glass-based triboelectric nanogenerators

Xin Xia [1,2,9], Ziqing Zhou[3,9], Yinghui Shang[3,4], Yong Yang [3,5,6] & Yunlong Zi [1,2,7,8]

Surface wear is a major hindrance in the solid/solid interface of triboelectric nanogenerators (TENG), severely affecting their output performance and stability. To reduce the mechanical input and surface wear, solid/liquid-interface alternatives have been investigated; however, charge generation capability is still lower than that in previously reported solid/solid-interface TENGs. Thus, achieving triboelectric interface with high surface charge generation capability and low surface wear remains a technological challenge. Here, we employ metallic glass as one triboelectric interface and show it can enhance the triboelectrification efficiency by up to 339.2%, with improved output performance. Through mechanical and electrical characterizations, we show that metallic glass presents a lower friction coefficient and better wear resistance, as compared with copper. Attributed to their low atomic density and the absence of grain boundaries, all samples show a higher triboelectrification efficiency than copper. Additionally, the devices demonstrate excellent humidity resistance. Under different gas pressures, we also show that metallic glass-based triboelectric nanogenerators can approach the theoretical limit of charge generation, exceeding that of Cu-based TENG by 35.2%. A peak power density of 15 MW·m$^{-2}$ is achieved. In short, this work demonstrates a humidity- and wear-resistant metallic glass-based TENG with high triboelectrification efficiency.

To address the growing energy demands, various sustainable energy harvesting technologies, such as solar cell[1,2], piezoelectric nanogenerators[3,4], thermoelectric nanogenerators[5,6] and triboelectric nanogenerators (TENGs)[7,8], etc., have been developed in recent years. Among all energy harvesters, TENGs become much popular in the field recently, because of the high output performance, broadly available materials, and flexible structure design. Based on the coupling effects of triboelectrification and electrostatic induction[9], the mechanical energy from environments, such as the wind[10], body motions[11], and ocean waves[12,13], etc., is converted into electricity by the TENG, which is able to power commercial load with the power management circuit[14]. Previous studies have demonstrated high surface charge density through triboelectrification by solid/solid interfaces[15,16]. However, the high wear rate between the solid pairs lead to the limited lifetime[17,18],

[1]Department of Mechanical and Automation Engineering, The Chinese University of Hong Kong, Shatin, N.T., Hong Kong SAR, China. [2]Thrust of Sustainable Energy and Environment, The Hong Kong University of Science and Technology (Guangzhou), Nansha, Guangzhou 511400 Guangdong, China. [3]Department of Mechanical Engineering, College of Engineering, City University of Hong Kong, Kowloon Tong, Kowloon, Hong Kong, China. [4]City University of Hong Kong (Dongguan), Dongguan 523000 Guangdong, China. [5]Department of Materials Science and Engineering, College of Engineering, City University of Hong Kong, Kowloon Tong, Kowloon, Hong Kong, China. [6]Department of Advanced Design and System Engineering, College of Engineering, City University of Hong Kong, Kowloon Tong, Kowloon, Hong Kong, China. [7]HKUST Shenzhen-Hong Kong Collaborative Innovation Research Institute, Futian, Shenzhen 518048 Guangdong, China. [8]HKUST Fok Ying Tung Research Institute, Guangzhou 511457 Guangdong, China. [9]These authors contributed equally: Xin Xia, Ziqing Zhou. ✉e-mail: yonyang@cityu.edu.hk; yunlongzi@gmail.com

hindering the development of TENG. Additionally, as limited by the triboelectrification efficiency of metal/polymer pairs, the output performance is still relatively low[19,20]. Nanoscale structures, such as nanofibers by electrospinning or pyramid arrays by photolithograph[21–23], are preferred to enhance the contact intimacy and thus the triboelectrification efficiency, while the friction and surface wear may become more severe, which affects the output performance and stability. Thus, alternative methods are still required to achieve high triboelectrification efficiency with low friction/wear.

To enhance the triboelectrification efficiency, the solid/liquid-based TENGs were proposed with various liquid types and device designs, achieving high output performance[12,24,25]. However, the solid/liquid based TENGs may be limited by the environment. The water source is required to ensure the long-term operation of most droplet-based TENGs, where the solid-liquid interactions were realized by the external environment, such as the rain or the ocean water[26,27]. Some designs with packaged liquids into the structure meet troubles in long-term stability of the performance, (e.g., water may be evaporated; the liquid metal may be oxidized quickly), limiting further application scopes. Additionally, the surface charge density by liquid/solid interfaces is usually limited as compared with that from solid/solid interfaces[12,28]. Therefore, it is essential to discover solid materials with similar advantages in liquids, such as low friction and high resistance in wear, while the high charge density can be still achieved. Previous studies have demonstrated that the metallic glass (MG) presents excellent performance serving as catalyst because of its high effective contact area[29,30], fast ion transfers and high stability[31,32], while studies on its energy harvesting performance were still lacking. Here, the MG reflects an amorphous structure in atomic level with no grain boundaries, which is similar to liquids and totally different from the common metals with ordered crystalline structures[31,33].

Herein, to enhance the triboelectrification efficiency in solid/solid interfaces with the high surface charge generation capability, MG with disordered atomic structures was employed as the triboelectric interface of TENG, demonstrating improved output performance compared with the copper (Cu) -based TENG due to low atomic density of MG samples. To investigate the electrical characteristics of MG-based TENGs, systematic experiments were conducted to evaluate the impact of normal load, MG sample types, gas pressure, relative humidity (RH), etc., on the output performance. 3 samples, including $Zr_{45}Cu_{40}Al_{15}$, $Zr_{45}Cu_{35}Al_{20}$, and $Zr_{50}Cu_{40}Al_{10}$, were focused in this work, with the Cu as the reference sample. As reflected by the mechanical characterizations, MG has a much better wear resistance than Cu, which is also reflected by the morphologies after the long-term test, indicating MG as an excellent wear-resistant and corrosion-resistant electrode for TENG. By investigating the relationship between the force input and the electrical output in both contact-separation (CS) and lateral sliding (LS) mode TENGs, better triboelectrification efficiency of MG is demonstrated compared with Cu, attributed to the low atomic density of MG and low friction coefficient. Additionally, because of the well hydrogen absorption capability of MG, the MG-based TENG demonstrates excellent humidity resistance. Finally, the capability of MG-based TENG for surface charge generation is evaluated in several scenarios, which approached the theoretical limit under different gas pressures. Besides, a peak power density of $15\,MW\,m^{-2}$ is achieved by a MG-based transistor-like TENG (T-TENG), successfully lightening 9 W LEDs. This work demonstrated the humidity-resistant, wear-resistant MG-based TENG with high performance, which is essential to push the output limit of TENG.

## Results
### Mechanical characterizations of metallic glass
Unlike the common metals with crystalline structures, MG presents a disordered atomic structure. It serves as structural materials and functional materials in various applications, attributed to its special mechanical properties and catalytic performance. MG ribbon, comprised of Zr, Cu, and Al, was fabricated by the melt-spinning process, as shown in Fig. 1a. Details of the fabrication are summarized in *Methods*. With different mixing ratio in each composition, the mechanical properties and the structure of MG sample are varied[34], and the MG samples selected in this work were guided by machine learning with well glass forming ability[35]. Here, the Zr-Cu-Al system is an excellent metallic-glass-former compared with other compositions and the Zr-based MG employed in this work reflected well glass forming capability and stability, nontoxicity, and low cost[36,37]. X-ray diffraction (XRD) patterns of $Zr_{45}Cu_{40}Al_{15}$, $Zr_{45}Cu_{35}Al_{20}$, and $Zr_{50}Cu_{40}Al_{10}$ are summarized in Fig. 1b–d, with the corresponding diffraction images shown as the insets, demonstrating the amorphous structures of the MG samples. High-resolution Transmission Electron Microscope (TEM) images to further confirm the amorphous phase are summarized in Supplementary Fig. 1. The Scanning Electron Microscopy (SEM) with Energy Dispersive X-Ray (EDX) analysis were carried out to characterize the elements distribution of our MG samples at the macroscopic scale, as shown Supplementary Fig. 2. According to these results, we conclude that the atoms are randomly distributed at the macroscopic scale, as illustrated in Fig. 1e(ii), while Cu has a crystalline atomic structure (Fig. 1ei). The compositions were listed in Supplementary Table 1 for reference. By employing the MG as the triboelectric interface of CS mode TENG, factors including normal force, atmosphere pressure, element components, and relative humidity were systematically studied to investigate the electrostatic properties of MG, as shown in Fig. 1f.

Figure 1g shows the photographs of the as-received MG ribbons with 5 mm in width and 20-40 μm in thickness, where the as-spun ribbon can be bended or twined, indicating a good mechanical deformability. SEM images for detailed surface information of MG samples as well as Cu are summarized in Supplementary Fig. 3–6, and the SEM images across the thickness were summarized in Supplementary Fig. 8. Additionally, the bottom surface of $Zr_{45}Cu_{40}Al_{15}$ shown in Supplementary Fig. 7 indicates that the bottom surface is a little bit rougher than the top one. This is because the bottom surface was in contact with the copper wheel during melting spinning, hence inheriting the surface roughness of the copper wheel. By comparison, the top surface appears brighter and smoother. In this work, the top surface was employed to investigate the output performance. The surface of Cu is much rougher and full of scratches, which might be attributed to different mechanical properties between Cu and MG. Atomic force microscope (AFM) images in Supplementary Fig. 9 illustrated the surface roughness of different samples, where MG demonstrated a smoother surface. Figure 1h illustrated the optical surface profiles of scratches after the sliding-mode indentation test of $Zr_{45}Cu_{40}Al_{15}$ (top) and Cu (bottom), which directly reflects the wear loss after the non-indentation experiments[38,39]. Details of the indentation experiments are described in *Methods*, and other profiles are summarized in Supplementary Fig. 10. Through scratch profiles in Supplementary Fig. 10 the wear coefficient of different samples was calculated and plotted in Fig. 1i. Here, Cu showed the worst wear loss, but MG samples demonstrated a much better wear resistance. The related parameters were summarized in Supplementary Table 2. As indicated in the previous study[40], MGs with a low atomic density tend to provide more electron donors than those with a high atomic density. To reveal the atomic density, we measured the mass density of different samples through *Drainage-method* and calculated the atomic density according to the molar density of the individual constituent elements, as shown in Fig. 1j, k and Supplementary Table 3. The details of the density measurement were summarized in *Methods*. To confirm the accuracy of the proposed method, the density of Cu was measured, which was similar to the theoretical value. The results demonstrated that all MG samples showed a much lower atomic density than Cu, and

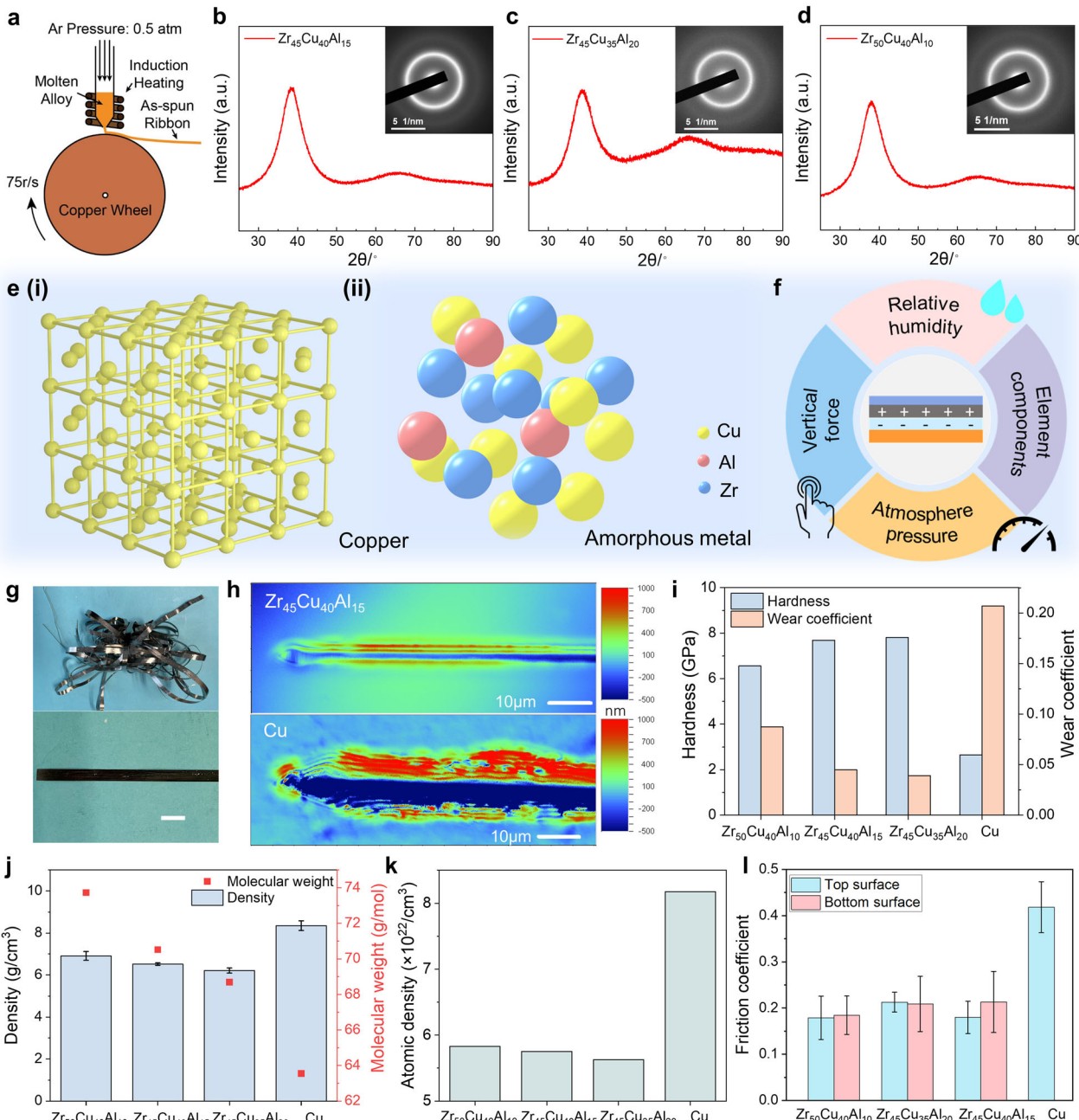

**Fig. 1 | Preparation and material characterization of the metallic glass material.**
**a** Fabrication of the metallic glass ribbon. X-ray diffraction patterns of
**b** $Zr_{45}Cu_{40}Al_{15}$, **c** $Zr_{45}Cu_{35}Al_{20}$ and **d** $Zr_{50}Cu_{40}Al_{10}$ and corresponding diffraction
images as insets. **e** Atomic structure of (**i**) Cu and (**ii**) metallic glass. **f** Factors for
evaluating the output performance of TENG based on metallic glass. **g** Photographs
of the as-received $Zr_{45}Cu_{40}Al_{15}$ ribbon. Scale bar = 1 cm. **h** Optical surface profiler
images of scratches after sliding-mode indentation of $Zr_{45}Cu_{40}Al_{15}$ (top) and Cu
(bottom). **i** Mechanical properties of different samples. **j** Density and **k** atomic
density of different samples. **l** Friction coefficient of different MG samples and Cu.
Error bar comes from the standard deviation of repeat experiments larger than 5
times. Source data are provided as a Source Data file.

$Zr_{50}Cu_{40}Al_{10}$ possessed the largest atomic density compared with the
other two MG samples.

The friction coefficient of Cu and both surfaces of MG was mea-
sured, where both surfaces are more slippery than Cu while the bottom
surface is a little bit rougher than the top surface, as shown in Fig. 1i.
Dynamic curves for measuring the fiction coefficient of different sur-
faces were shown in Supplementary Fig. 13, with related parameters in
Supplementary Table 4. Therefore, with the mechanical input and
durability considered, the low friction coefficient and the excellent
wear-resistant performance of MG make it an ideal material as the
triboelectric interface of TENG, and then the electrostatic properties

were investigated in detail. However, among all employed MG samples,
$Zr_{45}Cu_{35}Al_{20}$ is extremely brittle, which can be easily damaged under
shear force. The other two MGs are tough enough for the long-term
test and thus are preferred in further demonstrations. The SEM images
of scratches on each sample are summarized in Supplementary Fig. 13.
From the SEM images, shear bands can be observed around the
scratches without any signs of cracking, which demonstrates the
excellent deformability or mechanical stability of the MG samples.
Among the three samples, $Zr_{45}Cu_{40}Al_{15}$ exhibits the best strength
against shear flow as manifested by the much shallower shear band
traces.

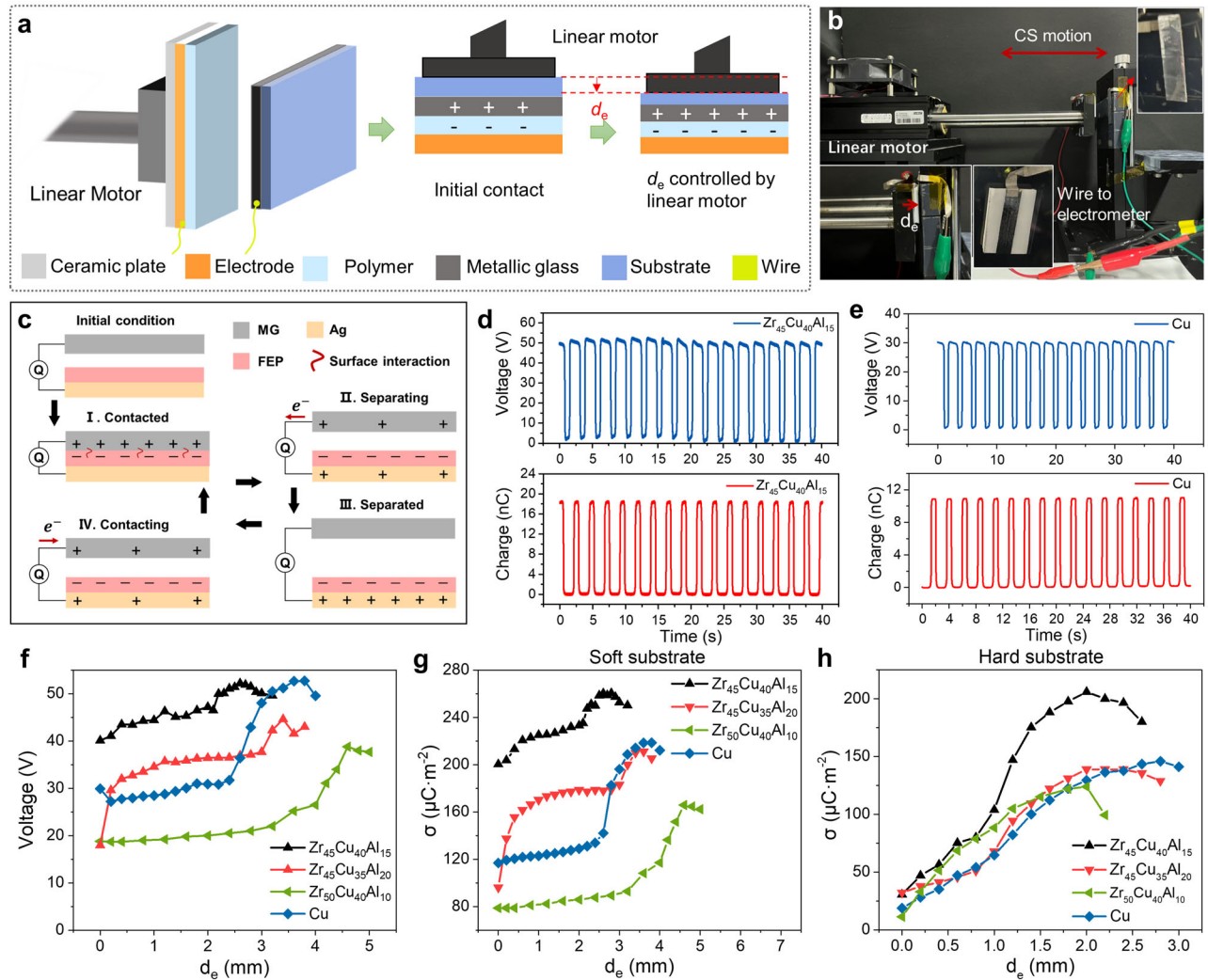

**Fig. 2 | Performance evaluation of CS mode TENG. a** Schematic diagram of the experimental setup. **b** Photograph of the device and experiment platform. **c** Operational mechanism of the MG-based TENG. Voltage (up) and charge (bottom) output of CS mode TENG with **d** Zr$_{45}$Cu$_{40}$Al$_{15}$ (5 mm × 15 mm) and **e** Cu (5 mm × 15 mm) as the metal contacting with FEP film. **f** $V_{OC}$ versus $d_e$ of different samples.

Relationship between surface charge density and $d_e$ of CS mode TENG with different metal materials under **g** soft substrate and **h** hard substrate. Here, the soft substrate was silicone foam tape with a thickness of 3 mm. Source data are provided as a Source Data file.

## Electrical performance of MG-based TENG

Limited by the surface roughness of primitive solid materials, air gap is inevitable in the solid/solid interface, resulting in insufficient contact intimacy and limited charge generation capability, thus a large mechanical load[41] or complicated surface nanopattern[42] is commonly employed to promote the effective contact. To quantitatively compare the electrical performance of different materials, we proposed the triboelectrification efficiency as a standard to reflect the practical charge generation capability, which is simply defined as:

$$\kappa = \frac{Q_{SC}}{AF_R} \qquad (1)$$

Where $\kappa$ is the standardized form of triboelectrification efficiency; $Q_{SC}$ is the recorded short-circuit charge output; A is the objective contact area; $F_R$ is the resistance force, as proportional to the mechanical energy waste during input. Unfortunately, $F_R$ is hard to be measured during experiments. Because $F_R$ is the vertical contact force in CS mode TENG, and $F_R = fN$ as the friction force where $f$ is the friction coefficient and $N$ is the vertical force in LS mode TENG, effect of vertical load on the electrical performance was studied separately in

both CS and LS mode TENGs, with the MG ribbon served as the interface contacting or sliding with polymer film. For the CS mode TENG, the vertical load was controlled by the moving displacement of the linear motor quantitatively. Fig. 2a illustrates the experiment platform for CS mode TENG. Based on the location sensor of the linear motor, the displacement corresponding to the initial contact was recorded first. The difference between displacements corresponding to full contact and initial contact was then defined as the excess displacement $d_e$, which was controlled by the linear motor. The $d_e$ only reflected the vertical load, where a large $d_e$ indicated a large vertical force because of the enhancement of work conducted by the linear motor. To avoid heavy impact with a large $d_e$, the acceleration and velocity of the linear motor was controlled at 0.1 m·s$^{-2}$ and 0.1 m·s$^{-1}$, respectively, resulting in a low motion frequency around 0.4 Hz. The photographs of setup as well as the electrode by MG are shown in Fig. 2b and Supplementary Fig. 14, where the negative charged side was fabricated by FEP layer. Assuming $N \propto d_e$ considering linear elastic deformation, the $\kappa$ by $d_e$ for CS mode TENG can be derived by:

$$\kappa_{d_e} = \frac{Q_{SC}}{AF_R} \sim \frac{Q_{SC}}{A \times d_e} \qquad (2)$$

Here, $\kappa_{d_e}$ is utilized to evaluate the charge generation capability of MG-based CS mode TENG under different $d_e$.

For the LS mode TENG, the normal force $N$ can be directly controlled by the weight. Thus, the $\kappa$ by force for the LS mode TENG can be calculated by:

$$\kappa_{\text{force}} = \frac{Q_{SC}}{AF_f} = \frac{Q_{SC}}{AfN} \sim \frac{Q_{SC}}{Af \times G} \qquad (3)$$

Where $G$ is the gravity of the loaded mass on the sliding layer of LS mode TENG.

The charge transfer processes in CS mode MG-based TENG are illustrated in Fig. 2c. Before the whole working processes, the triboelectric surfaces are separated without surface charge (initial condition). Firstly, the MG sample and the FEP layer contact each other, with surface charge generated on both surfaces by contact electrification. (I) Here, considering the amorphous structures at the atomic level of MG, the surface interaction between the FEP layer and the MG surface was marked to reflect the well charge generation capability. Then, the two surfaces are separated, and the electrons transfer from MG to the bottom electrode (Ag) to realize a new electrostatic equilibrium until the maximum separation displacement. (II-III) After that, the FEP layer moves close to the MG surface and the electrons transfer from the Ag to the MG until contact again. (IV) Then, a new working cycle starts. The open-circuit voltage ($V_{OC}$) and $Q_{SC}$ of different MG samples as well as Cu contacting with FEP layer at a certain $d_e$ of 2 mm were plotted in Fig. 2d, e and Supplementary Fig. 15, and the relationship between the electrical output and $d_e$ of MG samples, as well as, Cu were summarized in Fig. 2f. Here, the soft substrate at the electrode side was fabricated by a silicone foam tape with a thickness of 3 mm, as shown in Fig. 2b. Complete $V_{OC}$ and $Q_{SC}$ results under the soft substrate were summarized in Supplementary Fig. 16. To fully compare the electrical performance among different samples, the surface charge density from pure CE at different $d_e$ were calculated and summarized in Fig. 2g. Among all samples, $Zr_{50}Cu_{40}Al_{10}$ suggested a much lower surface charge density, but $Zr_{45}Cu_{40}Al_{15}$ was the best one with the maximized charge density up to 280 μC·m$^{-2}$. $Zr_{45}Cu_{35}Al_{20}$ presented a similar maximized output to Cu. When $d_e$ was relatively small, the output of $Zr_{45}Cu_{40}Al_{15}$ and $Zr_{45}Cu_{35}Al_{20}$ was much higher than Cu, demonstrating a better triboelectrification efficiency of the two MG samples. The results show that with the soft substrate, the output performance of different MG samples vary a lot due to the distinct atomic density as shown in Fig. 1k, where $Zr_{45}Cu_{40}Al_{15}$ demonstrated a lower atomic density while $Zr_{50}Cu_{40}Al_{10}$ is the highest. A lower atomic density brings a higher surface contact area and more electron donors[40], and then results in a higher surface charge density, which is consistent with experimental results in Fig. 2g. The worse output performance of $Zr_{45}Cu_{35}Al_{20}$ compared to $Zr_{45}Cu_{40}Al_{15}$ may result from its brittle structure, where cracking of $Zr_{45}Cu_{35}Al_{20}$ may happen during contact and then cause the surface charge dissipation. Therefore, by controlling the atomic density of MG samples through adjusting the MG components, the output performance of MG-based TENG can be manipulated. Additionally, Fig. 2g shows that, with the continuously increased $d_e$, the surface charge density increased slowly at the beginning, and then enhanced rapidly to the maximized output, and decreased eventually. Here, the rapid enhancement range with $d_e > 2$ mm may result from the tiny sliding motion between triboelectric surfaces due to the severe deformation of the soft substrate, and the sliding motion results in a better contact intimacy and then leads to a better charge generation capability, comparing to the pure contact electrification. Thus, to evaluate the effect of the deformation by the substrate, the relationship between the output performance and $d_e$ with the hard substrate was studied. The surface charge density with the hard substrate was summarized in Fig. 2h, where the output was lower than that with soft substrate considering the suppressed contact intimacy due to the limited $d_e$ and substrate deformation. With the hard substrate, the $V_{OC}$ and $Q_{SC}$ of the three MG samples and Cu depending on $d_e$ were plotted in Supplementary Fig. 17. Here, the $Zr_{45}Cu_{40}Al_{15}$ presented the best output performance, while output from other MG samples were similar to that from Cu. Thus, $Zr_{45}Cu_{40}Al_{15}$ always provided a better output performance in experiments, and thus it was preferred in further demonstrations in this work.

To further facilitate the triboelectric charge generation, the LS mode TENG was studied to confirm the improved output performance from MG, where the normal force was controlled by the load force $F_L$ or excess height $h_e$ of lifting elevator. The experimental setup was shown in the inset in Fig. 3a and results are summarized in Supplementary Fig. 18. Details of the operational processes are depicted in *Methods*. Here, the effect of $h_e$ was similar to that of $d_e$ for CS mode TENG, which provided a pre-tightening force between the triboelectric surfaces. The results demonstrated that a larger $h_e$ or weight resulted in higher surface charge density and $Zr_{45}Cu_{40}Al_{15}$ showed a better output performance than Cu with the same $h_e$, as shown in Fig. 3a, b. In addition, when $F_L$ was low, the surface charge density between the two samples was nearly the same, but the surface charge density of $Zr_{45}Cu_{40}Al_{15}$ under higher $F_L$ significantly exceeded that of Cu, up to 330 μC m$^{-2}$ in an atmosphere environment, which was much higher than that from the pure metal. (Fig. 3a) The better output performance of MG in both CS mode and LS mode TENG should be attributed to the better triboelectrification efficiency between MG and polymer surface because of the amorphous structures that brings larger specific surface area as well as the electron donors, which were also confirmed by the better catalytic performance of MG[30,31,40]. It should be noticed that in both CS mode and LS mode TENGs, the mechanical input with well-controlled force and displacement can hardly cause the severe plastic deformation that may induce fracture because the electrodes were fixed on the substrates through the Kapton tape serving as the "buffer" layer. The elastic strain by friction force, which was in ~‰ scale, was much slower than the elastic strain limit of MG samples (around 2.36%-2.47%), ensuring the stability of MG samples during the experiments. Details of calculation are depicted in Supplementary Note 1.

To better understand the results, the $\kappa$ of the aforementioned devices is calculated based on the experimental results. Figure 3c, d summarized $\kappa_{CS}$ for CS mode TENG with the hard substrate and soft substrate, respectively, where the zoom-in figure for that with the large $d_e$ with soft substrate is shown in the inset. The results demonstrated that all MG samples reflected a much better $\kappa$ compared with Cu by up to 339.2% ($d_e$ of 2.2 mm). Among all electrode materials, $Zr_{45}Cu_{40}Al_{15}$ demonstrated the best triboelectrification efficiency owing to its low friction coefficient and excellent mechanical deformability or stability. Similar results can be observed in the LS mode TENG, where $\kappa_{LS}$ of MG as the electrode as impacted by $F_L$ and $h_e$ are much higher than that of Cu by up to 184.3% ($F_L$ of 20 N) and 190.3% ($h_e$ of 2 unit), as shown in Fig. 3e, f.

The higher $\kappa$ from MG-based TENG may be due to the lower friction coefficient and low atomic density. As demonstrated in previous studies, the disturbed symmetry of MG former, led to remarkable enhancement in interfacial molecular bonding like hydrogen and diffusion of ions into sublayers, resulting in a higher volumetric absorption capacity[30–32]. Thus, the better interaction between the polymer chains and MG surface during the contact or sliding process resulted in a better charge generation capability of MG-based TENG. In addition, as illustrated in the definition of $\kappa$, the higher $\kappa$ suggests that a lower mechanical input is required for MG-based to realize a certain output performance, providing a promising potential for applications in various scenarios.

To highlight the contribution of triboelectrification efficiency on the enhancement of output performance, the surface charge density of different material pairs was compared in Fig. 4a. CS mode TENGs

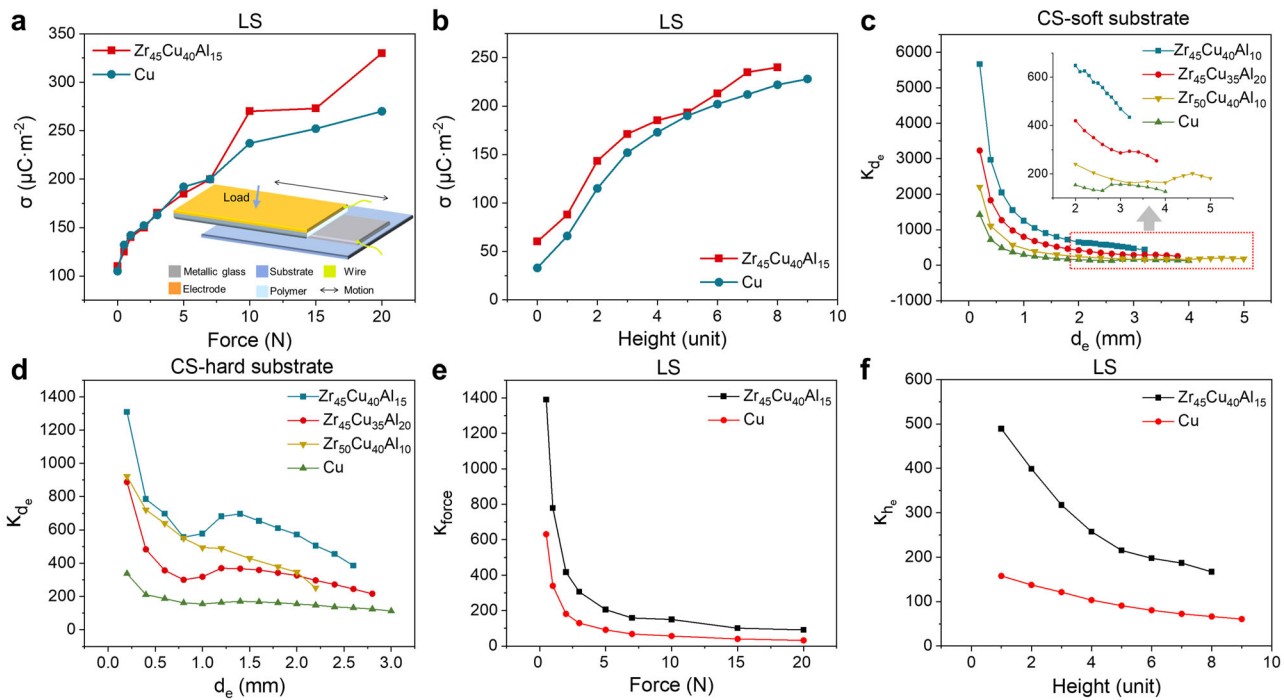

**Fig. 3 | Results of triboelectrification efficiency.** Surface charge density of LS mode TENG versus **a** force and **b** height. **c** $\kappa_{d_e}$ of CS mode TENG under soft substrate. **d** $\kappa_{d_e}$ of CS mode TENG under hard substrate. **e** $\kappa_{force}$ and **f** $\kappa_{h_e}$ of LS mode TENG. Source data are provided as a Source Data file.

with $Zr_{45}Cu_{40}Al_{15}$ or Cu contacting with different polymers, including FEP, PTFE, Kapton, Nylon-6, PET and PC, were studied. The results show that no matter $Zr_{45}Cu_{40}Al_{15}$ was positively charged or negatively charged, it always brought a higher surface charge density than that from Cu, demonstrating that the enhanced output performance of certain MG sample should be attributed to the improved triboelectrification efficiency rather than the polarity. The $Q_{SC}$ of $Zr_{45}Cu_{40}Al_{15}$ contacting with FEP and PC were plotted in Fig. 4b, while $Zr_{45}Cu_{40}Al_{15}$ was negatively charged after contacting with PC. The $Q_{SC}$ of other material pairs were plotted in Supplementary Fig. 20 for MG and Supplementary Fig. 21 for Cu. As control experiments, the output performance from polymer/polymer triboelectrification based TENG was measured, where MG or Cu was employed as the electrode for electrostatic induction only. Similar output performance from both devices with MG or Cu was recorded from PC/FEP SFT mode TENG, as shown in Supplementary Fig. 22, further confirming that the improved output was resulted from high triboelectrification efficiency of the MG/FEP interface. Models under atomic levels were proposed in Fig. 4c to illustrate the mechanism of the enhanced triboelectrification efficiency when MG with low atomic density was employed for triboelectrification, with charge transfer processes marked. Considering PTFE layer as the negatively charged materials, when the PTFE layer contacts with Cu, the effective contact between the PTFE chains and Cu is confined at the outer surface with limited electron donors due to the crystalline structure of Cu. However, due to the lack of grain boundaries and loose atomic packing in $Zr_{45}Cu_{40}Al_{15}$, the effective contact intimacy is enhanced a lot and more PTFE atoms are able to interact with MG atoms for charge generation, resulting a remarkable enhancement in output performance by MG-based TENG. Considering the electron affinity of different atoms, the effective interaction for charge transfer happens between endgroups of -F in PTFE and the nearby metallic atoms.

As demonstrated in previous studies, surface charge density of common TENGs decays a lot when the relative humidity (RH) of the surrounding environment is high. Considering the hydrogen absorption capacity of MG, the impact of RH on MG-based TENG may be less

compared with common TENGs. The effect of RH on the output performance of MG-based TENG was thus investigated. To control the RH of the environment, the CS mode TENG was placed in a sealed box, as shown in Fig. 4d, and a hygrometer was attached to reflect the in-situ RH. Molecular sieves were employed to dry the air and realize a low RH environment. The RH was enhanced by mist spray, and the $Q_{SC}$ was recorded after the RH became stable. Details of the experimental process are described in *Methods*. The evolution of surface charge at different RH of Cu/polymer pairs was studied first, as the reference. Figure 4e, f reflected the charge variation of Cu/FEP and Cu/PC versus RH, respectively, where the relative surface charge was defined as the ratio of recorded $Q_{SC}$ over the $Q_{SC}$ at the driest condition for comparison. As shown in the results, the relative surface charge decayed a lot when the RH continuously increased, when Cu was either negatively charged (Cu/FEP) or positively charged (Cu/PC). The relationship between the surface charge and RH for the MG-based TENG was illustrated in Fig. 4g–i. The corresponding $Q_{SC}$ and $V_{OC}$ at different RH were summarized in Supplementary Fig. 23. It can be noticed that, unlike the results from Cu/FEP, surface charge from MG/FEP pairs, including $Zr_{50}Cu_{40}Al_{10}$/FEP and $Zr_{45}Cu_{40}Al_{15}$/FEP, were much more stable with the increase of the RH, where larger than 80% surface charge was remained at the RH > 80%, especially for $Zr_{45}Cu_{40}Al_{15}$/FEP. However, the surface charge of $Zr_{45}Cu_{40}Al_{15}$/PC decayed faster with the increased RH, as compared with Cu/PC, while MG or Cu is negatively charged. The mechanism may be due to the better hydrogen absorption capacity of MG, which brought more hydrogen bonds on the surface. The hydrogen bonds facilitated the positive polarization of the surface, as demonstrated in previous studies[43], which then promoted the charge generation when MG was positively charged but hindered the performance when MG was negatively charged. Therefore, with the trade-off brought by the positive polarity enhancement from hydrogen bonds and charge decay from the moisture-gathering charges, the surface charge of MG/FEP pairs with MG as the positive side slightly increased and then decreased with the increase of the RH. In the meanwhile, MG/PC pair with MG as the negative side reflected a faster charge decay with the RH than that of Cu/PC. These results

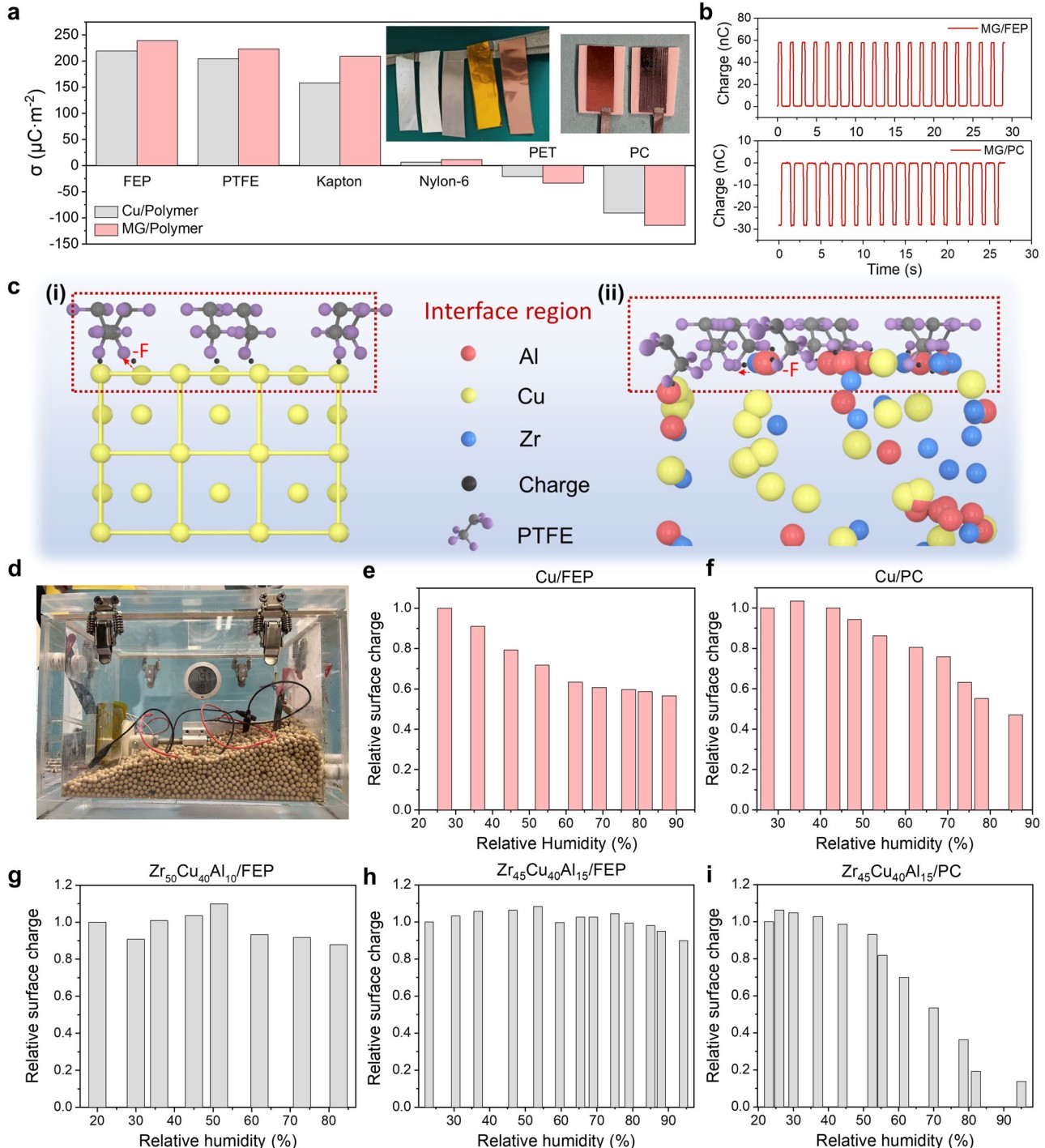

**Fig. 4 | Performance demonstration of the MG-based TENG. a** Comparison of surface charge density for the CS mode TENG with different material pairs, with photograph of the polymer sample shown in the inset. The size of electrode was 5mm-by-22mm. **b** Positive charge output of MG/FEP pair (top) and MG/PC pair (bottom). **c** Contact model under atomic level for (**i**) Cu/PTFE and (**ii**) MG/PTFE. **d** Photograph of the experimental platform for evaluating the effect of relative humidity. Relative surface charge of **e** Cu/FEP pair and **f** Cu/PC pair under different relative humidity. Relative surface charge of **g** $Zr_{50}Cu_{40}Al_{10}$/FEP pair, **h** $Zr_{45}Cu_{40}Al_{15}$/FEP pair and **i** $Zr_{45}Cu_{40}Al_{15}$/PC pair under different relative humidity. Source data are provided as a Source Data file.

demonstrated that besides with the high charge generation capacity, MG also exhibits excellent performance in RH resistance when it serves as the positive charged surface, greatly widening the application scopes in harsh environment. The charge transfer mechanism in RH-resistant ability of MG was explained by the potential well model as illustrated in Supplementary Fig. 24. The long-term durability tests were conducted with different samples, and the time duration was controlled as 11,000 seconds. The charge variation was summarized in

Supplementary Fig. 25 and Supplementary Table 5. It can be noticed that all samples demonstrated excellent charge stability during the long-term test, with similar reduction of surface charge (around 3nC after 3 h). After the durability test, the surface wear of different samples was investigated through optical microscopes. As shown in Supplementary Fig. 26, compared with MG, the Cu demonstrated a severer surface oxidation and discolored surface wrinkle/scratches after a long-time test. However, both $Zr_{50}Cu_{40}Al_{10}$ and $Zr_{45}Cu_{40}Al_{15}$ showed

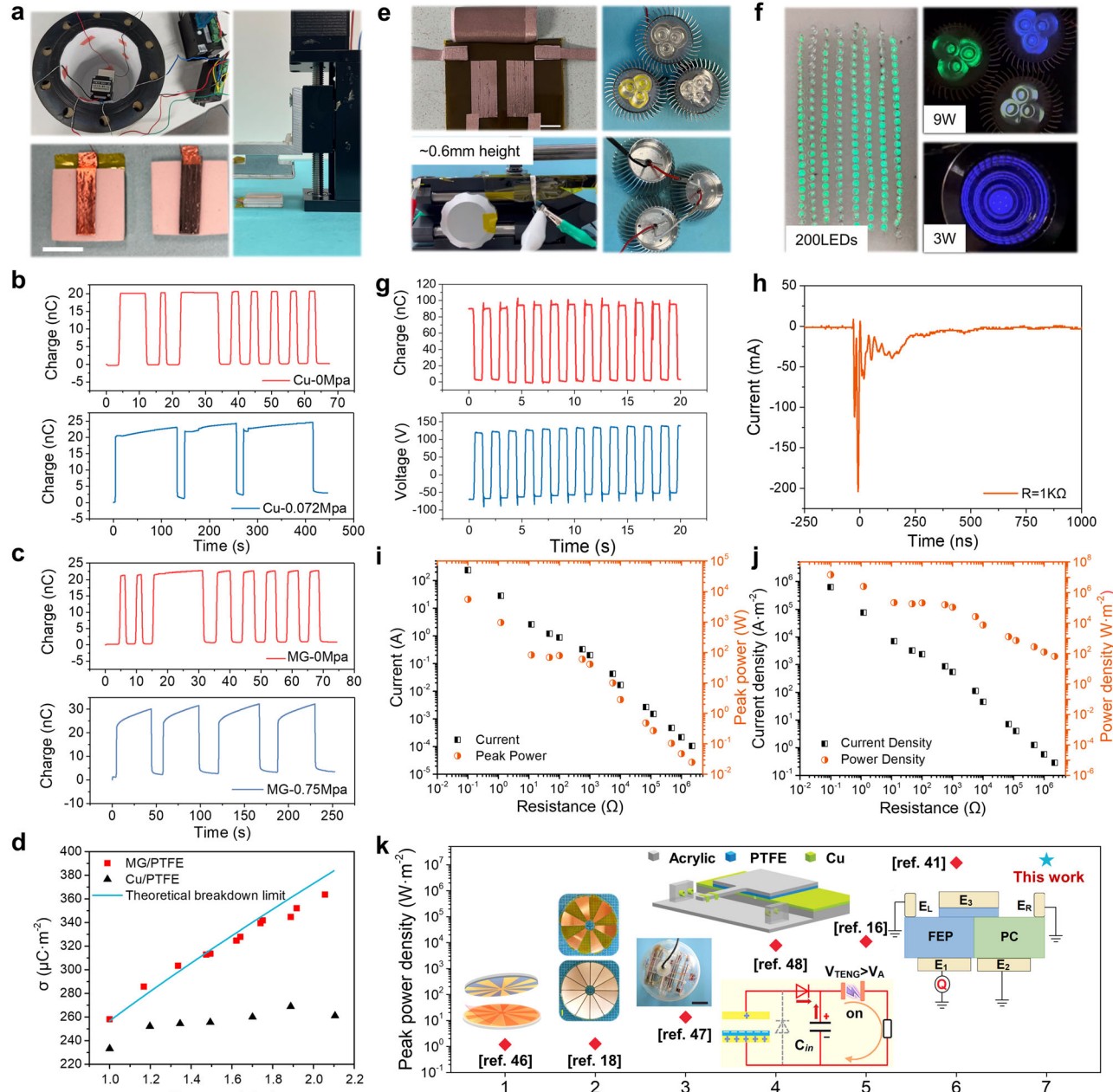

**Fig. 5 | Demonstration of the high-performance MG-based TENG. a** Photograph of experimental setup of the TENG with high gas pressure. Charge output at a different gas pressure of **b** Cu/PTFE pair and **c** MG/PTFE pair. **d** Summary of the surface charge density of different material pairs at different gas pressure, with the theoretical line compared. **e** Setup of the MG-based T-TENG. **f** Photograph for powered LEDs. **g** $Q_{SC}$ (top) and $V_{OC}$ (bottom) of MG-based T-TENG. **h** Current output of the device at resistance of 1kΩ. **i** The peak current and peak power, **j** the peak current density and peak power density depending on the load resistance.

Here, the MG sample is $Zr_{45}Cu_{40}Al_{15}$. The scale bar was 1 cm. **k** Comparison of peak power density of this work to other work. (Reference information: No.1: Reproduced with permission[46]. Copyright 2020, American Chemical Society. No.2: Reproduced with permission[18]. Copyright 2021, Springer Nature. No.3: Reproduced with permission[47]. Copyright 2020, Springer Nature. No.4: Reproduced with permission[48]. Copyright 2018, John Wiley & Sons. No.5: Reproduced with permission[16]. Copyright 2021, Elsevier. No.6: Reproduced with permission[41]. Copyright 2021, Springer Nature. Source data are provided as a Source Data file.

little difference in surface morphology, demonstrating a much better wear-resistance and durability as well as corrosion resistance. These results indicated that MG is a promising material for TENG in harsh environment with well durability.

### Demonstrations of MG-based TENG in different situations

To further confirm the excellent performance of MG-based TENG, applications in different scenarios were demonstrated with $Zr_{45}Cu_{40}Al_{15}$ selected as the triboelectric interface. Considering the remarkable triboelectrification efficiency of $Zr_{45}Cu_{40}Al_{15}$, it was tested

in the high-pressure $CO_2$ environment which can greatly suppress the breakdown discharge, to investigate the maximum charge density in the MG-based TENG[44]. The experiment platform was shown in Fig. 5a, where a chamber with a pressure gauge was utilized to realize the high-pressure condition. A small stepping motor was fixed in the chamber and the area of the electrode was around 5 mm × 18 mm. The $Q_{SC}$ under different gas pressure from Cu/PTFE and MG/PTFE were plotted in Fig. 5b, c, respectively, where higher enhancement was achieved in MG/PTFE pair. The relationship between the gas pressure and the surface charge density for the two material pairs was summarized in

Fig. 5d, with the theoretical breakdown limit plotted based on the pristine Paschen's law as the reference[19]. It can be noticed that the output charge of MG almost fit the theoretical limit perfectly, improving the charge density by 35.2% as compared with that of Cu, demonstrating the ability of MG-based TENG to effectively push the output performance until the breakdown limit. To demonstrate the energy output, a transistor-like TENG (T-TENG) was developed, where two switches were integrated in the structure to enhance the instantaneous power output, as demonstrated in previous studies[14,41,45]. Figure 5e shows the structures of the device and the experimental setup, where the height of the lifting elevator was carefully adjusted during the experiment, ensuring a small friction force. The schematic structure of the T-TENG was illustrated in Supplementary Fig. 27. The effective electrode size was around 15 mm × 25 mm only, which successfully powered 200 LEDs connected in series, as shown in Fig. 5f. More importantly, this small device was able to light a 9 W LED directly, even under the small mechanical input, indicating the high power output as well as the current output during the sliding process. Videos for powering the commercial loads were recorded in Supplementary Movie 1–4, and the system configuration was illustrated in Supplementary Fig. 28. The $V_{OC}$ and $Q_{SC}$ of the T-TENG were plotted in Fig. 5g, with $Q_{SC}$ of 100 nC (top) and $V_{OC}$ of around 210 V (bottom), indicating the surface charge density of 267 μC m$^{-2}$ was realized by the simple MG/FEP interfaces. The relationship between the electrical output and the load resistance was investigated, where the current output oscillated at high frequency was measured by an oscilloscope. Figure 5h show an extremely large current peak of around 200 mA was recorded at the resistance of 1 kΩ. The circuit for measuring the high-frequency current output was shown in Supplementary Fig. 29. Complete results of the peak current and peak power as well as the output densities depending on the load resistance were summarized in Fig. 5i, j. As shown in Fig. 5j, the maximized peak power density of around 15 MW·m$^{-2}$ was achieved by the small device, demonstrating the remarkable contact intimacy and power generation capability of Zr$_{45}$Cu$_{40}$Al$_{15}$ with a high triboelectrification efficiency. In this T-TENG, the charge accumulates when the switch is off and then the whole charge releases instantaneously when the switch is on, and thus the peak power decays with the increasing resistance, as discussed in Supplementary Note 2. As compared in Fig. 5k, the peak power density achieved by the MG/FEP pairs is beyond all previous studies under such a low vertical load, even without the charge promotion by opposite-charge-enhancement effect[41,46], further confirming the huge potential for certain MG samples to promote the output performance of TENG. Supplementary Movie 5 illustrated the small vertical load during measurement.

## Discussion

In summary, to take advantages of the high contact intimacy in solid/liquid interfaces and the high surface charge density in solid/solid TENG simultaneously, MG with amorphous atomic structures, including Zr$_{50}$Cu$_{40}$Al$_{10}$, Zr$_{45}$Cu$_{35}$Al$_{20}$ and Zr$_{45}$Cu$_{40}$Al$_{15}$, were employed as the triboelectric interface, with Cu as the reference sample. With the better wear resistance, MG suggested greatly improved triboelectrification efficiency compared with Cu by up to 339.2%, where Zr$_{45}$Cu$_{40}$Al$_{15}$ with lower atomic density demonstrated the best electrical output. This is because that when metals contact with polymers, more polymer atoms are able to closely interact with the atoms of MG in absence of grain boundaries and with lower atomic density for charge generation, while the contact between regular metals and polymers is restricted outside the crystalline structure with limited electron donors. The 3-hour long-term test of MG-based TENG demonstrated a stable output performance. More importantly, after the long-term test, the surface morphologies of all MG samples were nearly unchanged, while the Cu went through severe surface oxidation, wear, and wrinkle. This behavior indicates the MGs are excellent wear-resistant and corrosion-

resistant electrodes for TENG. Additionally, the MG/polymer with MG as the positive surface exhibited excellent RH-resistant ability, as compared with Cu. Finally, the capability of MG for surface charge generation to push the output limit of TENG was demonstrated in several scenarios, where the output charge nearly achieved the theoretical breakdown limit under high gas pressure, improving the charge density by 35.2%. Besides, a peak power density of 15 MW·m$^{-2}$ was realized by a MG-based T-TENG, with 9 W LEDs successfully lightened by the small device. This work may provide a new method to maximize the output performance of TENG, which is essential for widening the future applications of TENG.

## Methods

### Preparation for the MG material (melt spinning)

Pure metals, including Zr, Cu, and Al with a purity level higher than 99.95%, were used to prepare the MG ribbon samples. A lab-scale arc-melting furnace was employed to pre-melt the pure metals. The furnace was vacuumed to $8 \times 10^{-4}$ Pa first and then pumped up with Ar gas to 0.8 atm. The pure metals were melted to ingot for at least five times. A melted Ti ingot is applied to avoid oxidation. Then the alloy ingots are melted in a lab-scale induction-melting furnace with a vacuum as high as $8 \times 10^{-4}$ Pa and the ribbons are prepared by a single Cu roller to melt spinning with a rotating speed of 75 r/s, as shown in Fig. 1a. The as-received MG ribbons are around 5 mm in width and 20 μm-40 μm in thickness. The difference between the two surfaces results from the fabrication processes. During the melting spinning method, the bottom surface was in contact with the copper wheel during melting spinning, hence inheriting the surface roughness of the copper wheel. By comparison, the top surface appears brighter and smoother.

### Drainage-method for atomic density

Here the 99.97% ethyl alcohol was employed rather than water to reduce the air bubbles around samples during volume measurement. Before the density measurement, the density of ethyl alcohol was measured to remove the system error, where the mass of 5 ml ethyl alcohol was recorded through the balancing apparatus. For the density measurement of different samples, the mass was measured first, and then we recorded the mass of a 5 ml measuring cylinder with the sample. Then, we added ethyl alcohol into the measuring cylinder to 5 ml by an injector. By the mass difference, we can obtain the mass of ethyl alcohol and then we can calculate the corresponding volume of the sample. Finally, we got the density and the atomic density of different samples. To confirm the accuracy of the proposed method, the density of Cu was measured, which was similar to the theoretical value. Photographs of some related processes were shown in Supplementary Fig. 11.

### Fabrication of the involved devices in different experiments

For the MG-based CS mode and LS mode TENG, the as-received MG ribbons were directly pasted on a 100 μm-thick PET substrate, with the assistance of Kapton tape. The soft substrate involved in Figs. 2–3 was realized by the silicone foam tape with a thickness of around 3 mm, which was purchased from the *McMASTER-Carr*. The soft substrate in the experiments of gas pressure was a 1 mm thick foam tap, which was utilized to enhance the contact intimacy but weaken the deformation of the foam tape under different gas pressures. The hard substrate was realized by the PET substrate. The effective contact area of the devices was 5 mm in width, and 15 mm-20 mm in length, as marked in the corresponding figure legends. The electrode of metal/polymer TENG was fabricated by Cu tape pasting on the PET substrate, with the same contact areas. The polymer layers, including FEP and PTFE, Kapton, Nylon-6, PET, and PC with a thickness of 50 μm were also pasted on PET substrate by Kapton tape. The electrode on the back side of the polymer was silver or Cu with a thickness of 200 nm deposited by electron-beam evaporation.

The fabrication of T-TENG was a little bit different. The sharp side edge of the sample was smoothed by abrasive paper. Three $Zr_{45}Cu_{40}Al_{15}$ ribbons were aligned closely to form an electrode with width of 15 mm, a narrow Cu tape was employed at the bottom to ensure the conductivity among the three ribbons. A layer of 2 mm-thick PDMS was placed at the bottom to serve as the soft substrate. Then, a layer of Kapton tape was pasted on the PDMS to smooth the surface. Finally, two MG electrodes with size of 15 mm × 25 mm was pasted on the substrate through the Kapton tape, with the air gap between the two of 5 mm. Two switches with height of 1 mm were pasted on the substrate, which were close to the higher edge of the electrodes. The slider of the T-TENG was fabricated by a FEP layer pasted on the conductive fiber. A small area of the conductive fiber was exposed to air to achieve an automatic switch during periodical sliding motions. The structure of the whole device was shown in Fig. 5e and Supplementary Fig. 27. A layer of 2 mm-thick foam tape was pasted at the bottom of the slider layer.

In all involved TENGs, the mechanical input with well-controlled force and displacement can hardly cause the severe plastic deformation that may induce fracture because the electrodes were fixed on the substrates through the Kapton tape serving as the "buffer" layer.

### Experiment setup of excess displacement on CS mode TNEG

The electrode side was placed at the fixed side and the polymer layer was pasted at the linear motor side. Before measurement, the two surfaces were adjusted to exactly contact when the linear motor was switched off and no obvious deformation of the soft substrate. Then, based on the location sensor of the linear motor, the displacement under the initial contact condition was recorded first. The difference between displacement under over contact and that under initial contact was then considered as the excess displacement $d_e$, which was controlled by the linear motor. However, the $d_e$ was not equal to the deformation. It only reflected the vertical load, where a large $d_e$ indicated a large load. To avoid heavy impact when the $d_e$ was large, the acceleration and velocity of the linear motor were controlled at 0.1 m·s$^{-2}$ and 0.1 m·s$^{-1}$, resulting in a low motion frequency around 0.4 Hz. Before the measurement, the surface charge was removed by 99.97% ethyl alcohol. The electrical output by pure CE was then recorded during the periodical CS motions. The experiment platform for CS mode TENG was illustrated in Fig. 2a. The $Q_{SC}$ and $V_{OC}$ was recorded by the *Keithley 6514*.

### Experiment setup of vertical load in LS mode TNEG

Here, the silicone foam tap was attached at the bottom of MG as the substrate and PTFE layer was employed as the sliding layer because of its self-lubrication. The electrode side was placed at the lifting elevator and the polymer layer was pasted at the bottom of the slider. For the experiments depending on the height, the initial height (0 unit) was marked by adjusting the height to realize initial contact condition between the two surfaces. Here, a unit height was defined as an element of the controlling wheel of the lifting elevator, as shown Supplementary Fig. 19, and a unit was around 0.2 mm in height. Certain unit was adjusted by wheel element of the lifting elevator. Then, the $Q_{SC}$ and $V_{OC}$ at different height were measured under periodical sliding motion. For the experiments depending on the vertical load, the height was fixed around 2 unit, and the vertical force was loaded by the attached load. Before the measurement, the surface charge was removed by 99.97% ethyl alcohol as well. The experiment platform for LS mode TENG was illustrated by the inset in Fig. 3a and Supplementary Fig. 18a. The $Q_{SC}$ and $V_{OC}$ was recorded by the *Keithley 6514*.

### Experiment setup of evaluating the effect of RH

As shown in Fig. 4d, the CS mode TENG was placed in a sealed box, and a hygrometer was attached to reflect the in-situ RH. The electrode side was pasted on the wall of the box, and the polymer side was driven by a micro motor. Molecular sieves were employed to dry the air and realize a low RH environment. The molecular sieves were driest first through the oven with 70 °C. Before the measurement, the surface charge was removed by 99.97% ethyl alcohol. Then the initial surface charge was induced by sliding between the two surfaces. The measurement started when the RH was decreased to the lowest (around 20%). After each measurement, the two surfaces were held at contact condition, then the RH was enhanced by mist spray, and the $Q_{SC}$ was recorded after the RH became stable. The surface charge at different RH was extracted as the value when the $Q_{SC}$ was roughly stable. The $Q_{SC}$ and $V_{OC}$ was recorded by the *Keithley 6514*.

### Experiment setup of gas pressure

The soft substrate in the experiments of gas pressure was a 1 mm thick foam tap, which was utilized to enhance the contact intimacy but weaken the deformation of the foam tape under different gas pressures. The experiment platform was shown in Fig. 5a, where the reactor with pressure gage was utilized to realize the high-pressure condition. $CO_2$ was injected into the reactor to enhance the gas pressure. A small stepping motor was fixed in the reactor with the assistance of Kapton tape at the back side and bange tape at front side. The electrode side was fixed at the bottom while the PTFE layer was pasted on the micro-stepping motor. The working displacement was 1.5 cm fixed by the controller and the CS motion was controlled by hands through the manual mode of the controller. The area of the electrode was around 5 mm-by-18 mm. $Zr_{45}Cu_{40}Al_{15}$ and Cu were compared in this section. The initial output at air pressure was recorded first, then the gas pressure was enhanced slowly and the corresponding $Q_{SC}$ was measured. After the whole measurement, the $Q_{SC}$ after recovering air pressure was measured to check the fixation of the device because no window existed to monitor the condition by eyes. The $Q_{SC}$ and $V_{OC}$ was recorded by the *Keithley 6514*.

### Experiment setup of T-TENG

The electrode side of T-TENG was pasted on the lifting elevator and the slider was placed at the bottom of the slider. Before measurement, the working displacement was identified by contacting the switches at both sides, assisted by the location sensor of the linear motor. The height was controlled by the wheel element of the linear motor as well. The height was fixed at 0.6 mm (3 unit) during the experiments depending on the resistance. The $Q_{SC}$ and $V_{OC}$ was recorded by the *Keithley 6514*. The current with high frequency at different resistance was calculated by the voltage across the resistor. Here, the voltage was measured by an oscilloscope, the *Keysight DSOX2014A*, connected at both sides of the resistor. The circuit for measuring the voltage was illustrated in Supplementary Fig. 29.

### Mechanical characterization

The nanoindentation tests were subsequently performed using the *Hysitron TI950* nanoindentation system with a *Berkovich* tip, with photograph shown in Supplementary Fig. 12, which was calibrated using quartz standard. The hardness and elastic modulus were measured in sets of 9 indents with a loading of 8mN, and the dynamic friction coefficient was recorded with a loading of 8 mN as well. Scratch tests were performed at a speed of 2.5 μm/s and a constant normal force of 60 mN over a length of 100 μm on all test samples. For each sample after the scratch test, we applied SEM to observe the morphology of scratches and measure the surface profile along the scratch track by the optical surface profiler. By scanning the profile of the scratches, we obtained the relationship between the distance across the scratches and the height, which directly reflects the wear loss after the non-indentation experiments[38,39]. Then, the wear coefficient was calculated by the Archard wear equation:

$$K \sim \frac{HA_s}{P} \qquad (4)$$

Where, $K$ is the wear coefficient; $H$ is the surface hardness; $A_s$ is the wear loss and $P$ is the contact load.

## Data availability
Source data are provided with this paper.

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

## Acknowledgements

This work was funded by HKSAR the Research Grants Council General Research Funds (Grant no. 14200120, 14202121), National Natural Science Foundation of China (Grant no. 52275560), Guangdong Natural Science Funds for Distinguished Young Scholar (Grant no. 2023B1515020074), and the start-up fund of Hong Kong University of Science and Technology – Guangzhou (Grant no. G0101000092). This work was supported in part by the Project of Hetao Shenzhen-Hong Kong Science and Technology Innovation Cooperation Zone (HZQB-KCZYB–2020083). The work of Y.Y. is supported by the Research Grants Council (RGC), the Hong Kong government, through the General Research Fund (GRF) with the grant numbers CityU11200719 and CityU11213118 and through the NSCF-RGC joint research scheme with the grant number N_CityU 109/21.

## Author contributions

Y.Z., Y.Y., and X.X. conceived the idea. Z.Z. prepared the metallic glass samples. X.X. fabricated the TENGs in all related experiments. Z.Z. conducted the nanoindentation measurements and SEM. Y.S. conducted surface characterizations, including the nanoindentation measurements for wear coefficient and the optical profile images, AFM, TEM, SEM-EDX, and XRD, etc. X.X. conducted the experiments of electrical performance evaluations and compared the different performances of different MG samples and Cu. X.X. conducted optical microscopy for surface wear after durability. X.X. developed the mechanisms and repeated the main results. Z.Z. and Y.S. participated the data analysis. X.X. drafted the paper, Y.Z. and Y.Y. revised the paper. All authors participated in the interpretation of the data and production of the final paper.

## Competing interests

The authors declare no competing interests.
