## [Peer Review File · Nature Communications]

Metallic glass-based triboelectric nanogeneratorsREVIEWER COMMENTS

Reviewer #1 (Remarks to the Author):

The paper focuses on the replacement of a conventional Cu used as the triboelectric interface for TENG with metallic glass and the eventual performance enhancement. They have evaluated the impact of normal load, gas pressure, relative humidity (RH), etc., on different MG types on the output performance. Because of the favorable properties of MGs, I found it very inspiring to have them used in such triboelectric nanogenerators. Hence, the paper deserves to be published after minor revision: Compositions – This is an unorthodox way to define metallic glass compositions. Usually the total of the components should add up to 100%.

Fig 1c shows a slightly sharp peak on the second broad diffraction maximum. What is the reason for that?

Fig1e (ii) is certainly not intuitive enough to show the atomic clusters in the Zr-Cu-Al system. Better to have a closer-up simulation image

Fig 1h-j Nothing can be read from the scale bars, please make them significantly larger. Are there really pores, or rather casting defects or surface oxide?

Line 210 – I believe Figure 2j is misspelled

Line 248 $\mu\text{C m}^{-2}$, there is a typo in the minus sign

Fig 4 c (ii) I believe the interface region should be redrawn more realistically with more atoms in the bulk phase and a clear indication of which atoms interact with which endgroups of PTFE

Fig 5k Some inside figures are too small so it is impossible to read what is written there

Reviewer #2 (Remarks to the Author):

This paper presents a triboelectric nanogenerator with high wear-resistance and humidity resistance compared to ZrCuAl metallic glass with Cu. Improvements for peak density and humidity resistance are acknowledged, but I have several concerns to publish this paper in Nature Communication. Mainly, there is insufficient information on the experimental detail and analysis of material properties. Some discussion had been made with insufficient evidence that potentially lead misunderstanding. Another important point, although MG has high yield strength and hardness, it is very brittle because it does not have plastic deformation region. In the practical point of view, there is reliability concerns of MG although wear resistance that only derived from the hardness and modulus is improved. Detail comments are below.

1. Because there is size effect in indentation, detailed experimental conditions such as indentation depth, load, and materials of indenter tip etc. It is also necessary to check whether there is any size effect problem to compare different materials discussed in the paper.

2-1. While MG has a high yield strength, fracture occurs immediately over the point without plastic deformation. Deformation in the elastic region could be fine, but cracking may occur if it is outside the plastic region. The range of applied strain under several mechanical tests and modes in this paper should be described. Also the elastic strain range should be investigated of MG through micro-tensile test to understand mechanical stability with details.

2-2. As loading condition of indentation is a compression, cracking may be observed. Since the wear is related to shear force, I just wonder whether the wear resistance derived from hardness are still meaningful for metallic glass.

3. Labeling of the peak location is required in the XRD graph of Figure 1b-d, and adding a diffraction image seems to be useful for confirming amorphous phase.

4. Surface roughness may be changed by the process, and it is also possible to modify it through surface treatment. Comparing the performance of materials while maintaining the difference in surface roughness is required to characterize MG materials such as friction or electrical property. The effect of roughness and pure materials behavior need to be separated. In addition, the roughness of the surface is not measured, but only a qualitative comparison is made through images. It is desirable to

compare the properties between materials in a uniform state by polishing the surface.

5. Porosity is also pointed out as an important characteristic of MG. It is necessary to quantify porosity indirectly even through density measurement through methods such as XRR.

6. What is the rationale for the improvement in the humidity properties of the MGs due to the hydrogen absorption capacity of the MGs? There should be some reference or explanation why ZrCuAl MG has better capability of hydrogen absorption.

Reviewer #3 (Remarks to the Author):

In the manuscript entitled "Metallic-Glass toward High-Performance, Humidity-Resistant and Wear-Resistant Triboelectric Nanogenerator with Ultrahigh Peak Power Density" Xia et al. has utilized metallic glass (MG) as the triboelectric material in solid-solid contact based triboelectric nanogenerator in order to enhance the efficiency of contact electrification. The developed TENG demonstrates lower friction coefficient and higher wear resistance which attribute for enhanced triboelectric output. Moreover, the porous morphology as confirmed by FESEM and absence grain boundary in metallic glass further contribute to improve efficiency of solid-solid contact electrification. Besides, unlike conventional solid-solid contact electrification, MG based contact electrification demonstrates humidity independent triboelectric output owing to its hydrogen absorption property. Additionally, it is well observed that MG based TENG could also achieve the theoretical limit of charge generation by manipulating the gas pressure of ambience, which exceeds the charge generation of Cu based TENG. Finally, energy harvesting performance of the developed MG based TENG has been assessed in terms of peak power density where maximum dissipated peak power density has been determined as 15 MW·m⁻² with successful demonstration of lighting up 9 W LED. Although, the proposed strategy to enhance output power of solid-solid TENG as reported in this work is quite unique, the manuscript requires several fundamental revisions according to following remarks for the consideration to publish in a reputed journal paper such as Nature Communication.

1. In the introduction section, the authors have claimed that even though solid-liquid contact electrification provides less wear abrasion and friction coefficient, it still suffers from short term stability owing to atmospheric evaporation and oxidation of liquid. However, the claim is not valid for all modes of solid-liquid interactions. As an example, for droplet mode TENG with water as liquid triboelectric material, the evaporation of water for long-term operation will not affect the triboelectric performance as each droplet falling from a nozzle connected with water resource would interact with the solid surface separately. Moreover, there are number of publications demonstrating the long-term stability of solid-liquid based TENG especially in droplet mode. (ACS Appl. Mater. Interfaces 2020, 12, 31351–31359, ACS Nano 2021, 15, 18172–18181, Nano Energy 77 (2020) 105093).

2. The authors need to address the reasons for selecting Zr-based MG in this study. In Figure 1(I), the authors are asked to explain the differences between Surface 1 and Surface 2 because their cooling rates are different during melt spinning from solidification. Which Surface was used for TENG in this study? Use top/bottom surfaces throughout the manuscript so that these surfaces are not confused with Surface 1/Surface 2.

3. The morphological representation of metallic glass using SEM image is quite confusing as the authors have provided a tilted view of MG which is showing two different surfaces. In this regard, the authors must provide only top view of the MG surface for the better understanding of readers. For the confirmation of the thickness of MG, the authors must capture a proper cross-sectional SEM image of the structure along with the substrate.

4. In SEM images of manuscript and supporting document, there are no clear evidence of porous structure in MG ribbon. Particularly, the authors claimed the Zr_{4.5}Cu₄Al_{1.5} is the most porous structure among the three samples. Since the porous structure is one of key properties to achieve TENG performance, effects of porosity should be addressed and presented unambiguously. For instance, to increase porosity by chemical etching may be used to further enhance TENG property.

5. A neat schematic for explaining the operational mechanism of the proposed MG based TENG must be provided in Figure 2.

6. Photographic images in Figure 2b are not clear enough to represent the proposed device and experimental platform.

7. In Figure 5i, the characteristics curve of peak power output for different value of load resistance

looks quite different than that of previously published works as mentioned in Figure 5k. The authors must provide valid explanation regarding this peak power output curve otherwise they need to repeat the experiment to validate the characteristics of peak power with respect to different value of load resistance.

8. In Figure 5j, the unit mentioned for power density is wrong. It should be W/m².

9. In Figure S15, time duration to evaluate the durability of the proposed MG based TENG is not enough to prove its higher wear resistance and long-term energy harvesting capability as in the case of solid-liquid based TENG, the durability test has already been conducted for at least 3 hours (10800 sec.).

10. In Figure S15, time duration to evaluate the durability of the proposed MG based TENG is not same for all the triboelectric materials mentioned in Figure S15. Therefore, in order to compare the result of Cu, Zr_{4.5}Cu₄Al_{1.5} and Zr₅Cu₄Al₁ as the electrode contacting with FEP, the time duration for each must be same for proper evaluation.

11. It is well noted in Figure S15 that even all the MG cannot exhibit long-term durability such as Zr₅Cu₄Al₁. In this regard, the authors must provide proper explanation behind the reduction of surface charge of Zr₅Cu₄Al₁ compared to Zr_{4.5}Cu₄Al_{1.5}.

12. To facilitate the durability evaluation, the authors are asked to assess the MG and Cu (such as SEM images) after the durability tests are completed.

Reviewer #4 (Remarks to the Author):

In this paper, the authors demonstrated the wear-resistant, humidity-resistant TENG with high triboelectrification efficiency by using the metallic-glass (MG) as the triboelectric electrode. The material the used is new for the TENG field and the output performance by it is attractive, which may push the output limit of TENG. More importantly, this paper developed the TENG with ultrahigh peak power density that may break the record among previous studies. Therefore, I would recommend this manuscript for the possible publication in Nature Communications after minor clarifications. The comments to the author are given as below:

1. As shown in Figure 2f and g, the relationship between the surface charge density and the excess displacement is interesting. Why is there a faster enhancement in surface charge density when the excess displacement is larger than 2 mm for all samples when the soft substrate was employed?
2. As reported by the authors the MG ribbon was fabricated by the melt spinning method, why the two surfaces of the MG ribbon reflect a different morphology? One of it seems like brighter and smoother, and the friction coefficient is different as well.
3. The authors concluded that the better output performance of the MG-based TENG was attributed to higher triboelectrification efficiency between the MG/polymer interfaces. Why? Please clarify it clearer.
4. It was noted that the electrode in performance demonstration was very narrow in width, resulting a very small device. Why is it so small? Limited by the fabrication process? If so, as an energy-harvesting technology, will it limit the further applications of the MG-based TENG?
5. As summarized in Figure 5k, the authors reported that the extremely high peak power output by the MG-based TENG may serve as the new record. However, it can be noted that the enhancement between this work and the previous one (ref. 36) is not so significant, so what's the advantage of this work compared with the previous one?

Some typos are required to be revised, as listed below:

1. The mark (e) in the figure legends of figure 3 was missed, please revise it.
2. What do you mean by the step in Figure 2a, the excess displacement? Please define and revise it. So does the axis in Figure S6-7, please check it.

REVIEWER COMMENTS

Reviewer #1 (Remarks to the Author):

The paper focuses on the replacement of a conventional Cu used as the triboelectric interface for TENG with metallic glass and the eventual performance enhancement. They have evaluated the impact of normal load, gas pressure, relative humidity (RH), etc., on different MG types on the output performance. Because of the favorable properties of MGs, I found it very inspiring to have them used in such triboelectric nanogenerators. Hence, the paper deserves to be published after minor revision:

Answer:

We like to express our sincere thanks to the reviewer for clearly understanding the significance, innovation, and broad impact of this work.

Compositions – This is an unorthodox way to define metallic glass compositions. Usually the total of the components should add up to 100%.

Answer:

Thank the reviewer for the suggestion.

We have double checked that the sum of subscript in different samples should be 100. Thus, we revised the words used in the manuscript, and the samples were changed to $Zr_{50}Cu_{40}Al_{10}$, $Zr_{45}Cu_{40}Al_{15}$ and $Zr_{45}Cu_{35}Al_{20}$. All related labels in figures have been revised as well.

Fig 1c shows a slightly sharp peak on the second broad diffraction maximum. What is the reason for that?

Answer:

Thank the reviewer for the consideration.

As amorphous metals, no sharp peak should exist in the XRD results. The origination of the slightly sharp peak on the second broad diffraction maximum from $Zr_{45}Cu_{35}Al_{20}$ is unclear, which may result from contaminations. To confirm the results, we repeated the XRD measurements, and the new results are summarized in **Figure R1**, where no sharp peak exist in all results, demonstrating the non-crystalline structures of these MG samples.

Figure R1. X-ray diffraction results. (a) $Zr_{45}Cu_{40}Al_{15}$. (b) $Zr_{45}Cu_{35}Al_{20}$. (c) $Zr_{50}Cu_{40}Al_{10}$.

Fig1e (ii) is certainly not intuitive enough to show the atomic clusters in the Zr-Cu-Al system. Better to have a closer-up simulation image

Answer:

Thank the reviewer for the suggestion. The nature of atomic clusters (short range order) is a very interesting topic. In order to fully understand the details, we need further experiments, such as EXAFS, STEM and etc, which however has gone beyond the scope of the work (here we are more focused on the macroscopic behavior and properties).

With that being said, we carried out SEM-EDX experiments to characterize the elemental distribution at the macroscopic scale, as shown **Figure R3**. According to these results, we conclude that the atoms are randomly distributed at the macroscopic scale, as illustrated in **Figure R2**. We also list the compositions in **Table R1** for reference.

Figure R2. Zoom-in schematic atomic structure of $Zr_{45}Cu_{40}Al_{15}$. Atom number of 20.

Figure R3. SEM-EDX results for atom distributions (a) $Zr_{45}Cu_{40}Al_{15}$; (b) $Zr_{50}Cu_{40}Al_{10}$; (c) $Zr_{45}Cu_{35}Al_{20}$.

Table R1. Atom compositions of different samples from SEM-EDX results.

Samples	Zr	Cu	Al
$Zr_{45}Cu_{40}Al_{15}$	46.4%	37.8%	15.8%
$Zr_{50}Cu_{40}Al_{10}$	51.6%	38%	10.4%
$Zr_{45}Cu_{35}Al_{20}$	46.5%	33.1%	20.4%

Fig 1h-j Nothing can be read from the scale bars, please make them significantly larger. Are there really pores, or rather casting defects or surface oxide?

Answer:

Thank the reviewer for the suggestion.

We have revised **Figure. 1h-j** so that the scale bars are shown clearly. We are sorry that by “porous” we actually mean low atomic density. As indicated in previous study,¹ the MG material with low atomic density provides more electron donors than that with high atomic density. To reveal the atomic density, we measured the mass density of different samples through *Drainage-method* and calculated the atomic density according to the components, as shown in **Figure R4** and **Table R2**. Here the 99.07% ethyl alcohol was employed rather than water to reduce the air bubbles around samples during volume measurement. The results show that $Zr_{45}Cu_{40}Al_{15}$ reflected the lower atomic density compared with $Zr_{50}Cu_{40}Al_{10}$, consistent with previous conclusions. The details of the density measurement were summarized in *Methods*. To confirm the accuracy of the proposed method, the density of Cu was measured, which was similar to the theoretical value. The results also

demonstrated that all MG samples show a much lower atomic density than Cu, consistent with the conclusion in RH-resistant measurements.

Figure 1 (h-j). Scanning electron microscopy (SEM) images of (h) $Zr_{45}Cu_{35}Al_{20}$, (i) $Zr_{50}Cu_{40}Al_{10}$ and (j) $Zr_{45}Cu_{40}Al_{15}$.

Figure R4. Comparison of (a) density and (b) atomic density of different samples.

Table R2. Atomic density of different samples.

Samples	Molecular weight (g/mol)			Density (g/cm ³)	Atomic density ($\times 10^{22}/cm^3$)
$Zr_{50}Cu_{40}Al_{10}$	73.72855			6.910239404	5.83104048
$Zr_{45}Cu_{40}Al_{15}$	70.51643			6.521007979	5.75324807
$Zr_{45}Cu_{35}Al_{20}$	68.6882			6.214908483	5.62912973
Cu	63.546			8.351060367	8.17602216
Atomic weight (g/mol)	Zr	Cu	Al	Theoretical density of Cu: 8.96 g/cm ³ ; Measured density of ethyl alcohol: 0.79422 g/cm ³ (theoretical: 0.7893 g/cm ³ at 20°C).	
	91.224	63.546	26.9815		

Line 210 – I believe Figure 2j is misspelled

Answer:

Thank the reviewer for the suggestion.
The related description has been revised.

Line 248 μC m-2, there is a typo in the minus sign

Answer:

Thank the reviewer for the suggestion.
The related description has been revised.

Fig 4 c (ii) I believe the interface region should be redrawn more realistically with more atoms in

the bulk phase and a clear indication of which atoms interact with which endgroups of PTFE

Answer:

Thank the reviewer for the suggestion.

The purpose of **Figure 4c** (ii) was to schematically illustrate the different contact intimacy for the charge enhancement in MG-based TENG.

To provide a clearer contact picture, the charge transfer processes were marked. Considering the electron affinity of different atoms, the effective interaction for charge transfer happens between endgroups of -F in PTFE and the nearby metallic atoms, as shown in the revised interface diagram in **Figure R5**.

Figure R5. Revised contact model under atomic level for (i) Cu/PTFE and (ii) MG/PTFE.

Fig 5k Some inside figures are too small so it is impossible to read what is written there

Answer:

Thanks for the reviewer's consideration.

The related figures have been revised.

Reviewer #2 (Remarks to the Author):

This paper presents a triboelectric nanogenerator with high wear-resistance and humidity resistance compared to ZrCuAl metallic glass with Cu. Improvements for peak density and humidity resistance are acknowledged, but I have several concerns to publish this paper in Nature Communication. Mainly, there is insufficient information on the experimental detail and analysis of material properties. Some discussion had been made with insufficient evidence that potentially lead misunderstanding. Another important point, although MG has high yield strength and hardness, it is very brittle because it does not have plastic deformation region. In the practical point of view, there is reliability concerns of MG although wear resistance that only derived from the hardness and modulus is improved. Detail comments are below.

Answer:

We like to express our sincere thanks to the reviewer for considerations and suggestions of this work.

1. Because there is size effect in indentation, detailed experimental conditions such as indentation

depth, load, and materials of indenter tip etc. It is also necessary to check whether there is any size effect problem to compare different materials discussed in the paper.

Answer:

Thank the reviewer for the suggestion.

The detailed experimental conditions of the indentation measurement were detailed in *Methods*.

In this manuscript, we aim at demonstrating the better electrical output performance of MG-based TENG compared with the one with metal as the electrode, thus the samples with as-received conditions were employed in the fabrication. To avoid the size effect in the electrical measurement, the size of all devices for performance comparison were maintained almost the same (5mm in width and around 20mm in length).

2-1. While MG has a high yield strength, fracture occurs immediately over the point without plastic deformation. Deformation in the elastic region could be fine, but cracking may occur if it is outside the plastic region. The range of applied strain under several mechanical tests and modes in this paper should be described. Also the elastic strain range should be investigated of MG through micro-tensile test to understand mechanical stability with details.

Answer:

We thank the reviewer for the consideration.

As discussed in the manuscript, the MG samples were employed as the electrodes, as shown in the device photographs in **Figure R6c**. In both contact-separation mode (**Figure R6a**) and sliding mode (**Figure R6b**) TENGs, the mechanical input with well-controlled force and displacement can hardly cause the severe plastic deformation that may induce fracture because the electrodes were fixed on the substrates through the Kapton tape serving as the “buffer” layer. With such the buffer layer providing additional elastic range, we didn’t observe fractures in tests. For the sliding mode, the averaged maximized vertical load on the MG sample in **Figure R6b** was around $\frac{20N}{30mm \times 25mm} \times 5mm \times 28mm = 3.73N$, and thus the maximized force of friction is around 0.75N, (friction coefficient of 0.2) which can be used to estimate the shear force. Thus, the elastic strain can be calculated by:

$$\varepsilon = \frac{F}{AE_r} \approx \frac{0.75N}{5mm \times 20\mu m \times 100GPa} = 0.75\text{‰}.$$

Similarly, for the CS mode TENG, the vertical load can be calculated by $F = \frac{m\Delta V}{\Delta t}$.

Considering the maximized velocity of 0.1m/s, the impulse of the linear motor was around 0.3kg m/s. Here, the mass of the slider of motor is around 3kg. The time interval was selected from the charge output under d_c of 3.2 mm, as 0.2s (**Figure R6f**). Thus, the maximized vertical load of CS mode TENG was around 1.5N. Then, during the relative sliding induced by the vertical load, the shear force was estimated at 0.3N (friction coefficient of 0.2) and the related strain was 0.3‰. Here, the friction of coefficient and the modulus were obtained by the indentation test, as indicated in

Figure 11 and **Table R3**. Therefore, the elastic strain range employed in this work was smaller than 1‰ for all modes, which is small enough to avoid the fractures during working.

Additionally, as we mentioned in the manuscript, among the three samples, only $Zr_{45}Cu_{35}Al_{20}$ is very brittle, which can be easily fractured by external forces. (**Figure R6d**) Therefore, it was not utilized in further demonstrations in this manuscript. The other two samples demonstrate better mechanical properties, where the as-spun ribbon can be bended or twined, as shown in **Figure R6e**, which is stable enough for our tests, especially under such a low strain range.

To further demonstrate the mechanical stability, we conducted a long-term durability test, with the results shown in **Figure R7**. It can be noticed that, after 3-hour test (more than 10k cycles), the metallic glass maintains well mechanical structures as well as electrical performance. Therefore, the mechanical stability of the MG-based TENG is convincing.

Figure R6. Mechanical illustrations. (a) Setup of CS mode TENG. (b) Setup of LS mode TENG. (c) Photographs of electrodes. Photographs of (d) $Zr_{45}Cu_{35}Al_{20}$ and (e) $Zr_{45}Cu_{40}Al_{15}$. (f) Time interval of the CS mode TENG under d_e of 3.2mm. Here the electrode was $Zr_{45}Cu_{40}Al_{15}$.

Figure R7. Durability test. Long-term (11000s) charge measurement of (a) Cu/FEP; (b) $Zr_{45}Cu_{40}Al_{15}$ /FEP and (c) $Zr_{50}Cu_{40}Al_{10}$ /FEP pairs under soft substrate of 1 mm. Here the sample size is 5 mm \times 25 mm and the excess displacement is 1 mm. The size of the FEP layer is 1 cm \times 3.5 cm.

2-2. As loading condition of indentation is a compression, cracking may be observed. Since the wear is related to shear force, I just wonder whether the wear resistance derived from hardness are still meaningful for metallic glass

Answer:

We thank the reviewer for the consideration.

As we mentioned, the MG samples were directly pasted on the substrate as the electrodes, and because of the fixation of the Kapton tape as the buffer layer and well-controlled force/displacement, no structural fracture as well as cracking observed in the $Zr_{45}Cu_{40}Al_{15}$ and $Zr_{50}Cu_{40}Al_{10}$ during operation. Besides, during our performance demonstration, the contact-separation mode TENG was preferred, where the vertical load was the dominant mechanical input. Therefore, we think the wear resistance derived from the hardness are meaningful for metallic glass.

To better illustrate the wear resistance, we conducted the indentation experiment with sliding mode, where the nano-probe with 60mN load sliding on the sample surface, resulting in a contact pressure at \sim GPa level. Then, the scratches on samples were analyzed through SEM and Optical Surface Profiler. As shown in the SEM images in **Figure R8**, shear bands can be noticed around the scratches, and no cracking was observed after the testing, demonstrating the mechanical stability of the MG samples. Among the three samples, $Zr_{45}Cu_{40}Al_{15}$ demonstrated the best shear strength

because of slighter shear bands.

Figure R8. SEM images of the scratches on surface after indentation measurement. (a) $Zr_{45}Cu_{40}Al_{15}$; (b) $Zr_{45}Cu_{35}Al_{20}$; (c) $Zr_{50}Cu_{40}Al_{10}$.

By scanning the profile of the scratches, we obtained the relationship between the distance across the scratches and the height in **Figure R9a** and **b**, which directly reflects the wear loss after the non-indentation experiments.^{2,3} **Figure R9b** shows MG samples reflected similar wear profile, while Cu suffered a severe wear loss. The related parameters were summarized in **Table R3**. Here, the wear coefficient was calculated by the Archard wear equation:

$$K \sim \frac{HA_s}{P}$$

Where, K is the wear coefficient; H is the surface hardness; A_s is the wear loss and P is the contact load. From **Figure R9c**, Cu has shown the largest wear coefficient, and MG samples demonstrated a much better wear resistance, consistent with our previous conclusions.

Figure R9. Analysis of wear resistance. (a) Optical surface profiler images of different samples. (b) Profile images of scratch. (c) Mechanical properties of different samples.

Table R3. Mechanical properties of different samples.

	Cu	Zr ₄₅ Cu ₄₀ Al ₁₅	Zr ₄₅ Cu ₃₅ Al ₂₀	Zr ₅₀ Cu ₄₀ Al ₁₀
As (µm ²)	4.6851	0.301	0.35085	0.79832
H (GPa)	2.65	7.81	7.69	6.56
E (GPa)	142.57	107.16	110.33	91.99
K	0.20693	0.03918	0.04497	0.08728

3. Labeling of the peak location is required in the XRD graph of Figure 1b-d, and adding a diffraction image seems to be useful for confirming amorphous phase.

Answer:

We thank the reviewer for the suggestion.

Because of the non-crystalline structure of metallic glass, there are no sharp peaks in the XRD results, only the broad Bragg diffraction peak, so the labeling was not added in the results. The XRD graphs and the diffraction image as well as the TEM images in **Figure R10** confirmed the amorphous phase of the MG samples employed in this manuscript, where no grain boundaries can be observed.

Figure R10. Confirmation of amorphous phase. XRD, diffraction image and high-resolution TEM image of (a-c) $Zr_{45}Cu_{40}Al_{15}$, (d-f) $Zr_{45}Cu_{35}Al_{20}$ and (g-i) $Zr_{50}Cu_{40}Al_{10}$.

4. Surface roughness may be changed by the process, and it is also possible to modify it through surface treatment. Comparing the performance of materials while maintaining the difference in surface roughness is required to characterize MG materials such as friction or electrical property. The effect of roughness and pure materials behavior need to be separated. In addition, the roughness of the surface is not measured, but only a qualitative comparison is made through images. It is desirable to compare the properties between materials in a uniform state by polishing the surface.

Answer:

We thank the reviewer for the consideration.

In this manuscript, we aim at demonstrating the better electrical output performance of MG-based TENG, and thus the samples employed in the fabrication were in the as-received condition, without surface treatment such as polishing. The surface roughness may be changed after long-term measurement by the friction, and the change from each sample may be different, especially the Cu electrode with lower hardness than MG, as shown in **Table R3**, which is very complicated. It is hard to maintain the same surface roughness during periodic contact-separation or sliding motions for each sample, so we did not consider the surface roughness change. However, we investigated the long-term electrical performance of TENGs with different electrode materials and recorded the surface morphology change by optical microscope, which reflected a more practical performance evolution during working. As shown in **Figure R11**, both $Zr_{50}Cu_{40}Al_{10}$ and $Zr_{45}Cu_{40}Al_{15}$ showed little change in surface morphology while Cu suffers a severe surface wear and oxidation.

Additionally, the surface roughness of different samples under as-received condition was measured by atomic force microscopy (AFM), as shown in **Figure R12**. Here, the scanning size for each sample is $8\mu\text{m} \times 8\mu\text{m}$. It can be noticed that all MG samples reflect a lower surface roughness

compared to Cu, consistent with our previous conclusions. Additionally, among all MG samples, $Zr_{50}Cu_{40}Al_{10}$ shows the largest surface roughness, while the other two are similar.

Figure R11. Surface morphologies change before and after long-term durability test of (a) Cu, (b) $Zr_{50}Cu_{40}Al_{10}$ and (c) $Zr_{45}Cu_{40}Al_{15}$. Optical microscope results, scale bar of $100\mu m$.

Figure R12. AFM images for surface roughness. The scanning size for each sample is $8\mu m \times 8\mu m$.

5. Porosity is also pointed out as an important characteristic of MG. It is necessary to quantify

porosity indirectly even through density measurement through methods such as XRR.

Answer:

Thank the reviewer for the suggestion. Unfortunately, because of the limitation of the surface roughness, the XRR cannot be employed to the MG samples, where a mirror surface is required by this method. Thus, we measured the density through the traditional drainage method.

We are sorry that by “porous” we actually mean low atomic density. As indicated in previous study,¹ the MG material with low atomic density provides more electron donors than that with high atomic density. To reveal the atomic density, we measured the mass density of different samples through *Drainage-method* and calculated the atomic density according the components, as shown in **Figure R13** and **Table R4**. Here the 99.07% ethyl alcohol was employed rather than water to reduce the air bubbles around samples during volume measurement. The results show that $Zr_{45}Cu_{40}Al_{15}$ reflected the lower atomic density compared with $Zr_{50}Cu_{40}Al_{10}$, consistent with previous conclusions. The details of the density measurement were summarized in *Methods*. To confirm the accuracy of the proposed method, the density of Cu was measured, which was similar to the theoretical value. The results also demonstrated that all MG samples show a much lower atomic density than Cu, consistent with the conclusion in RH-resistant measurements.

Figure R13. Comparison of (a) density and (b) atomic density of different samples.

Table R4. Atomic density of different samples.

Samples	Molecular weight (g/mol)			Density (g/cm ³)	Atomic density (x10 ²² /cm ³)
Zr ₅₀ Cu ₄₀ Al ₁₀	73.72855			6.910239404	5.83104048
Zr ₄₅ Cu ₄₀ Al ₁₅	70.51643			6.521007979	5.75324807
Zr ₄₅ Cu ₃₅ Al ₂₀	68.6882			6.214908483	5.62912973
Cu	63.546			8.351060367	8.17602216
Atomic weight (g/mol)	Zr	Cu	Al	Theoretical density of Cu: 8.96g/cm ³ ; Measured density of ethyl alcohol: 0.79422g/cm ³ (theoretical: 0.7893g/cm ³ at 20°C).	
	91.224	63.546	26.9815		

6. What is the rationale for the improvement in the humidity properties of the MGs due to the hydrogen absorption capacity of the MGs? There should be some reference or explanation why ZrCuAl MG has better capability of hydrogen absorption.

Answer:

Thank the reviewer for the consideration.

As demonstrated in previous studies, water is usually relatively positive due to triboelectric series,^{4, 5, 6} and it was also demonstrated that the absorption of hydrogen (the generation of the hydrogen bonds) on the triboelectric surface can enhance the positive triboelectric polarity⁷. Therefore, the hydrogen absorption capacity of the MG enhances the positive triboelectric polarity, which compensates the charge dissipation brought by the RH at the beginning. However, when the RH is continuously enhanced, the free water molecule will cause further charge dissipation of TENG, which will eventually decrease the output, as shown by the results from Cu/polymer TENGs as well as the previous results. **(Figure 4)**

Such the improved absorption of hydrogen is due to more active sites and faster electron transfer ability in MG compared with other metals, which can be also demonstrated by its performance on catalyzing hydrogen evolution reaction: DOI: 10.1038/NMAT2542;⁸ ACS Appl. Energy Mater. 2018, 1, 2630–2646;⁹ Adv. Mater. 2016, 28, 10293–10297^{10, 11}

Reviewer #3 (Remarks to the Author):

In the manuscript entitled “Metallic-Glass toward High-Performance, Humidity-Resistant and Wear-Resistant Triboelectric Nanogenerator with Ultrahigh Peak Power Density” Xia et al. has utilized metallic glass (MG) as the triboelectric material in solid-solid contact based triboelectric nanogenerator in order to enhance the efficiency of contact electrification. The developed TENG demonstrates lower friction coefficient and higher wear resistance which attribute for enhanced triboelectric output. Moreover, the porous morphology as confirmed by FESEM and absence grain boundary in metallic glass further contribute to improve efficiency of solid-solid contact electrification. Besides, unlike conventional solid-solid contact electrification, MG based contact electrification demonstrates humidity independent triboelectric output owing to its hydrogen absorption property. Additionally, it is well observed that MG based TENG could also achieve the theoretical limit of charge generation by manipulating the gas pressure of ambience, which exceeds the charge generation of Cu based TENG. Finally, energy harvesting performance of the developed MG based TENG has been assessed in terms of peak power density where maximum dissipated peak power density has been determined as 15 MW·m⁻² with successful demonstration of lighting up 9 W LED. Although, the proposed strategy to enhance output power of solid-solid TENG as reported in this work is quite unique, the manuscript requires several fundamental revisions according to following remarks for the consideration to publish in a reputed journal paper such as Nature Communication.

Answer:

We sincerely thank the reviewer for clearly understanding the significance, innovation and broad impact of this work.

1. In the introduction section, the authors have claimed that even though solid-liquid contact electrification provides less wear abrasion and friction coefficient, it still suffers from short term stability owing to atmospheric evaporation and oxidation of liquid. However, the claim is not valid for all modes of solid-liquid interactions. As an example, for droplet mode TENG with water as liquid triboelectric material, the evaporation of water for long-term operation will not affect the triboelectric performance as each droplet falling from a nozzle connected with water resource would interact with the solid surface separately. Moreover, there are number of publications demonstrating the long-term stability of solid-liquid based TENG especially in droplet mode. (ACS Appl. Mater. Interfaces 2020, 12, 31351–31359, ACS Nano 2021, 15, 18172–18181, Nano Energy 77 (2020) 105093).

Answer:

Thank the reviewer for providing the related literatures.

We would admire that the limitations were not valid for all modes of solid-liquid interactions. We actually mean the solid-liquid based TENGs are limited by the environment. As you mentioned, the water source is required to ensure the long-term working of most droplet-based TENGs, where the solid-liquid interactions were realized by the external environment, such as the rain or the ocean water, which may be limited by the working environment.^{12,13} For example, water droplets or ocean wave were required to directly contact with the device.¹⁴ To make the idea clearer, we have revised the claim in the introduction as “*To enhance the triboelectrification efficiency, the solid/liquid based TENGs was proposed with various liquid types and device designs, achieving high output performance.*^{5, 6, 15} However, the solid/liquid based TENGs may be limited by the environment. The water source is required to ensure the long-term operation of most droplet-based TENGs, where the solid-liquid interactions were realized by the external environment, such as the rain or the ocean water.^{12,13} Some designs with packaged liquids in the structure meet troubles in long-term stability of the performance, (e.g., water may be evaporated; the liquid metal may be oxidized quickly), limiting further application scopes.”

2. The authors need to address the reasons for selecting Zr-based MG in this study. In Figure 1(l), the authors are asked to explain the differences between Surface 1 and Surface 2 because their cooling rates are different during melt spinning from solidification. Which Surface was used for TENG in this study? Use top/bottom surfaces throughout the manuscript so that these surfaces are not confused with Surface 1/Surface 2.

Answer:

Thank the reviewer for the suggestions.

First of all, the Zr-Cu-Al is a kind of excellent metallic-glass-former. Compared with other composition, the Zr-based MG employed in this work reflected well glass generation capability and stability, nontoxicity and low cost.^{16, 17}

The difference between the two surfaces results from the fabrication processes. During the melting spinning method, the bottom surface was in contact with the copper wheel during melting

spinning, hence inheriting the surface roughness of the copper wheel. By comparison, the top surface appears brighter and smoother.

The manuscript was revised according to the suggestions and the top surface is used in TENG because it is smoother compared with the bottom surface with less local defects.

3. The morphological representation of metallic glass using SEM image is quite confusing as the authors have provided a tilted view of MG which is showing two different surfaces. In this regard, the authors must provide only top view of the MG surface for the better understanding of readers. For the confirmation of the thickness of MG, the authors must capture a proper cross-sectional SEM image of the structure along with the substrate.

Answer:

Thank the reviewer for the suggestion.

Here the SEMs in Figure S1 was employed to reflect the different side edge with sharing bands as well as the cross-sectional structures. To provide a clearer surface morphology, the top view of different MG materials was provided in **Figure R14-R16** to show the smooth surface. And the cross-sectional SEM image of as-received MG samples are shown in **Figure R17**, where $Zr_{50}Cu_{40}Al_{10}$ reflected a less smooth surface across the thickness.

Figure R14. SEM images of the top surface of $Zr_{45}Cu_{35}Al_{20}$ with different scales.

Figure R15. SEM images of the top surface of $Zr_{45}Cu_{45}Al_{15}$ with different scales.

Figure R16. SEM images of the top surface of $Zr_{50}Cu_{40}Al_{10}$ with different scales.

Figure R17. SEM images across the thickness of different samples. (a) $Zr_{45}Cu_{40}Al_{15}$; (b) $Zr_{45}Cu_{35}Al_{20}$; (c) $Zr_{50}Cu_{40}Al_{10}$.

4. In SEM images of manuscript and supporting document, there are no clear evidence of porous structure in MG ribbon. Particularly, the authors claimed the $Zr_{4.5}Cu_4Al_{1.5}$ is the most porous structure among the three samples. Since the porous structure is one of key properties to achieve TENG performance, effects of porosity should be addressed and presented unambiguously. For instance, to increase porosity by chemical etching may be used to further enhance TENG property.

Answer:

Thank the reviewer for the suggestions.

We are sorry that by “porous” we actually mean low atomic density. As indicated in previous study,¹ the MG material with low atomic density provides more electron donors than that with high atomic density. To reveal the atomic density, we measured the mass density of different samples through *Drainage-method* and calculated the atomic density according the components, as shown in **Figure R18** and **Table R5**. Here the 99.07% ethyl alcohol was employed rather than water to reduce the air bubbles around samples during volume measurement. The results show that $Zr_{45}Cu_{40}Al_{15}$ reflected the lower atomic density compared with $Zr_{50}Cu_{40}Al_{10}$, consistent with previous conclusions. The details of the density measurement were summarized in *Methods*. To confirm the accuracy of the proposed method, the density of Cu was measured, which was similar to the theoretical value. The results also demonstrated that all MG samples show a much lower atomic density than Cu, consistent with the conclusion in RH-resistant measurements.

Figure R18. Comparison of (a) density and (b) atomic density of different samples.

Table R5. Atomic density of different samples.

Samples	Molecular weight (g/mol)			Density (g/cm ³)	Atomic density ($\times 10^{22}/cm^3$)
$Zr_{50}Cu_{40}Al_{10}$	73.72855			6.910239404	5.83104048
$Zr_{45}Cu_{40}Al_{15}$	70.51643			6.521007979	5.75324807
$Zr_{45}Cu_{35}Al_{20}$	68.6882			6.214908483	5.62912973
Cu	63.546			8.351060367	8.17602216
Atomic weight (g/mol)	Zr	Cu	Al	Theoretical density of Cu: 8.96 g/cm ³ ; Measured density of ethyl alcohol: 0.79422 g/cm ³ (theoretical: 0.7893 g/cm ³ at 20°C).	
	91.224	63.546	26.9815		

5. A neat schematic for explaining the operational mechanism of the proposed MG based TENG must be provided in Figure 2.

Answer:

Thank the reviewer for the suggestions.

We added the operational mechanism of the MG-based TENG, which is similar to the normal TENG devices. Here, the CS mode MG-based TENG is illustrated as an example, and the charge transfer processes are summarized in the following figure (**Figure R19**). Before the whole working processes, both triboelectric surfaces are separated without surface charge (initial condition). Firstly, the MG sample and the FEP layer contact each other, with surface charge generated on both surfaces by the contact electrification. (I) Here, considering the amorphous structures at the atomic level of MG, the surface interaction between the FEP layer and the MG surface was marked to reflect the well charge generation capability. Then, the two surfaces are separated, and the electrons transfer from MG to the bottom electrode (Ag) to realize a new electrostatic equilibrium until the maximum separation displacement. (II-III) After that, the FEP layer moves close to the MG surface and the electrons transfer from the Ag to the MG until contact again. (IV) Then, a new working cycle starts.

Figure R19. Operational mechanism of the MG-based TENG.

6. Photographic images in Figure 2b are not clear enough to represent the proposed device and experimental platform.

Answer:

Thanks for the suggestions.

The photograph has been replaced to reflect the proposed device.

Figure R20. Photograph of the platform setup.

7. In Figure 5i, the characteristics curve of peak power output for different value of load resistance looks quite different than that of previously published works as mentioned in Figure 5k. The authors must provide valid explanation regarding this peak power output curve otherwise they need to repeat the experiment to validate the characteristics of peak power with respect to different value of load resistance.

Answer:

We thank the reviewer for the consideration.

For TENG in most previous studies, the peak power of TENG usually increased first and then decreased with the resistance continuously increasing, resulting a maximized peak power appeared at a large matching impedance (larger than 10MΩ). However, by our design of transistor-like TENG in **Figure 5e**, the peak power decays with the increasing resistance. The different peak power output curve among our work and previously published works resulted from the integrated switches in our work. Generally, charge accumulation happens along with power output during the periodic motions in a common TENG in previous studies.¹⁸ When a TENG directly connects to a resistor, the voltage across the load varies with the load, and a maximized peak power exists at the matched impedance resistance. However, with the integrated switches on both sides in our design, the structure is under short-circuit condition at starting and ending of each working cycle, as shown in **Figure R21**. During periodic motions of the transistor-like TENG, charge accumulates when the switch is off and then the whole charge releases instantaneously when the switch is on. Thus, when the resistance R increases, the peak power output decreases as $P_{\text{peak}} = V^2/R$, resulting a totally different characteristics compared with previous studies.

Figure R21. Schematic structure of T-TENG.

To further confirm the output characteristics, the experiments were repeated, where the peak power decreased with the increasing resistance, as shown in **Figure R22**. Here, a maximized peak power density around 8.34MW/m^2 was realized in the experiment at the RH around 68%, further confirming the output performance of the MG-based T-TENG. As the $P=V^2/R$, a slight voltage change may result in a significant change in peak power, and thus the peak power density was changed. The decreasing in peak power density may result in the different slider as well as the higher RH, but the output characteristics are the same in the repeated experiments with well output performance.

Figure R22. Resistance impedance of T-TENG. Relative humidity around 68%.

8. In Figure 5j, the unit mentioned for power density is wrong. It should be W/m^2 .

Answer:

Thank the reviewer for the suggestions.

The manuscript has been revised.

9. In Figure S15, time duration to evaluate the durability of the proposed MG based TENG is not enough to prove its higher wear resistance and long-term energy harvesting capability as in the case of solid-liquid based TENG, the durability test has already been conducted for at least 3 hours (10800 sec.).

Answer:

Thank the reviewer for the suggestion.

The long-term durability tests for 11000 sec. were conducted with different samples. The results are summarized as follows, where all pairs demonstrated well long-term durability.

Figure R23. Durability test. Long-term (11000s) charge measurement of (a) Cu/FEP; (b) $Zr_{45}Cu_{40}Al_{15}$ /FEP and (c) $Zr_{50}Cu_{40}Al_{10}$ /FEP pairs under soft substrate of 1 mm. Here the sample size is 5 mm × 25 mm and the excess displacement is 1 mm. The size of the FEP layer is 1 cm × 3.5 cm.

10. In Figure S15, time duration to evaluate the durability of the proposed MG based TENG is not same for all the triboelectric materials mentioned in Figure S15. Therefore, in order to compare the result of Cu, $Zr_{4.5}Cu_{4}Al_{1.5}$ and $Zr_{5}Cu_{4}Al_{1}$ as the electrode contacting with FEP, the time duration for each must be same for proper evaluation.

Answer:

Thank the reviewer for the suggestion.

The long-term durability tests were conducted with different samples, and the time duration was controlled as 11000 sec. (larger than 3 hours) for all samples. Results are shown in **Figure R23**.

11. It is well noted in Figure S15 that even all the MG cannot exhibit long-term durability such as $Zr_{5}Cu_{4}Al_{1}$. In this regard, the authors must provide proper explanation behind the reduction of surface charge of $Zr_{5}Cu_{4}Al_{1}$ compared to $Zr_{4.5}Cu_{4}Al_{1.5}$.

Answer:

Thank the reviewer for the consideration.

The charge decay during the long-term durability test of three samples is summarized in **Table**

R6. It can be noticed that all samples demonstrated excellent charge stability during the long-term test, with similar reduction of surface charge (around 3nC after 3 hours). Generally, triboelectric charge itself reflects slight fluctuation, which may be related to the surrounding environment. Here, all charge reduction is less than 10%, and the difference among the three samples is only 1% ~ 2%. Thus, the output can be considered as stable. Such a slight charge reduction was considered as stable in previous studies as well.^{19, 20, 21}

Table R6. Durability test of different samples.

Sample	Initial charge (nC)	Final charge (nC)	Charge reduction ΔQ (nC)	Decay ratio
Cu	33	30.1	2.9	8.788%
Zr ₄₅ Cu ₄₀ Al ₁₅	37.4	34.5	2.9	7.754%
Zr ₅₀ Cu ₄₀ Al ₁₀	30.2	27.2	3	9.934%

12. To facilitate the durability evaluation, the authors are asked to assess the MG and Cu (such as SEM images) after the durability tests are completed.

Answer:

Thank the reviewer for the consideration.

In fact, for the CS mode TENG, the surface wear was not obvious, but the MG demonstrated a better long-term surface stability. To clearly observe the surface wear, the optical microscope images were summarized in **Figure R24**. As shown in these optical microscope images, compared with MG, the Cu demonstrated a severer surface oxidation and discolored surface wrinkle/scratches after a long-time test. However, both Zr₅₀Cu₄₀Al₁₀ and Zr₄₅Cu₄₀Al₁₅ showed little difference in surface morphology, demonstrating a much better wear-resistance and durability, as compared with Cu.

Figure R24. Surface morphologies change before and after long-term durability test of (a) Cu, (b) Zr₅₀Cu₄₀Al₁₀ and (c) Zr₄₅Cu₄₀Al₁₅. Optical microscope results, scale bar of 100 μ m.

Reviewer #4 (Remarks to the Author):

In this paper, the authors demonstrated the wear-resistant, humidity-resistant TENG with high triboelectrification efficiency by using the metallic-glass (MG) as the triboelectric electrode. The material the used is new for the TENG field and the output performance by it is attractive, which may push the output limit of TENG. More importantly, this paper developed the TENG with ultrahigh peak power density that may break the record among previous studies. Therefore, I would recommend this manuscript for the possible publication in Nature Communications after minor clarifications. The comments to the author are given as below:

Answer:

We sincerely thank the reviewer for clearly understanding the significance, innovation and broad impact of this work.

1. As shown in Figure 2f and g, the relationship between the surface charge density and the excess displacement is interesting. Why is there a faster enhancement in surface charge density when the excess displacement is larger than 2 mm for all samples when the soft substrate was employed?

Answer:

Thank the reviewer for the consideration.

As depicted in the manuscript, the soft substrate employed in this experiment is a 3-mm-thick silicone foam tape. When the excess displacement is very large (larger than 2 mm), the severe deformation of the soft substrate may cause slight relative sliding motion between both triboelectric surfaces, leading to the sudden enhancement in surface charge density. This is because the sliding motion results in a better contact intimacy and then leads to a better charge generation capability, comparing to the pure contact electrification. When the experiments were conducted under hard substrate, the relative sliding motion was avoided and the sudden enhancement cannot be observed, as shown in **Figure 2h**, further confirming that the faster enhancement originated from the relative sliding.

2. As reported by the authors the MG ribbon was fabricated by the melt spinning method, why the two surfaces of the MG ribbon reflect a different morphology? One of it seems like brighter and smoother, and the friction coefficient is different as well.

Answer:

Thank the reviewer for the consideration.

The difference between the two surfaces results from the fabrication processes. During the melting spinning method, the bottom surface was in contact with the copper wheel during melting spinning, hence inheriting the surface roughness of the copper wheel. By comparison, the top surface appears brighter and smoother.

3. The authors concluded that the better output performance of the MG-based TENG was attributed to higher triboelectrification efficiency between the MG/polymer interfaces. Why? Please clarify it clearer.

Answer:

Thank the reviewer for the consideration.

As illustrated by figure 4a, when MG was rubbed with different polymer pairs, MG/ Polymer pairs always demonstrated higher surface charge density no matter the MG was positively or negatively charged (figure 4b). This indicates that the high output was not related to the triboelectric polarization, otherwise only either positively or negatively charge MG would demonstrate a higher charge output. In addition, when MG was employed as only an induction electrode for dielectric/dielectric CS TENG without MG/dielectric contact, there is no obvious output enhancement compared to Cu, as shown in Figure S12, indicating that the excellent charge generation performance mainly results from the high triboelectrification efficiency of the MG/FEP interface.

4. It was noted that the electrode in performance demonstration was very narrow in width, resulting

a very small device. Why is it so small? Limited by the fabrication process? If so, as an energy-harvesting technology, will it limit the further applications of the MG-based TENG?

Answer:

Thanks very much for the suggestions.

Limited by the material fabrication instrument, especially the glass-forming ability, so the width of the MG sample is usually narrow. As an energy-harvesting technology, the power density or the charge density is the key parameter to evaluate the device performance. Although the size of the MG-based TENG is small, the charge density is remarkable, which can nearly achieve the output limit as calculated by theories as demonstrated in this manuscript, which in fact, is beneficial to the miniaturization of the TENG. Especially, as demonstrated by the T-TENG, the device with an electrode size of 1.5 cm-by-2.5 cm can directly power the 200LEDs connected in series as well as the 9 W LEDs, further confirming the excellent output performance of MG-based TENG. We believe the width of the MG sample may be increased by improving fabrication technology. On the other hand, the miniaturized devices with high charge density may be beneficial for system integration and the scale-up energy harvesting for irregular mechanical energy.

5. As summarized in Figure 5k, the authors reported that the extremely high peak power output by the MG-based TENG may serve as the new record. However, it can be noted that the enhancement between this work and the previous one (ref. 36) is not so significant, so what's the advantage of this work compared with the previous one?

Answer:

Thanks very much for the concerns.

We admit that the peak power enhancement by the MG-based TENG is not so significant compared with the one in Ref. [36],²² and the characteristic we emphasize about the metallic glass is the high triboelectrification efficiency. However, as discussed in the manuscript, unlike the previous studies, the high peak power density in this work was directly realized by the as received MG/FEP pair, without any other structure design or material treatment. The vertical load (the preloaded height by the lifting elevator) in the performance demonstration was very low, which was reflected in the video S5. In the meanwhile, the work in Ref. [36] emphasized the opposite-charge-enhancement effect to enhance the final output, where the device structure design is more complicated. The triboelectrification efficiency of the work in Ref. [36] should be worse than our work as reflected by the heavy mass load on the slider, which means the force input required to trigger TENG in our work is much lower to achieve similar level of output.

Some typos are required to be revised, as listed below:

1. The mark (e) in the figure legends of figure 3 was missed, please revise it.

Answer:

Thanks very much for the suggestions.

The manuscript has been revised based on the suggestions.

2. What do you mean by the step in Figure 2a, the excess displacement? Please define and revise it. So does the axis in Figure S6-7, please check it.

Answer:

Thanks very much for the suggestions.

The manuscript has been revised based on the suggestions.

References for response only:

1. Jia Z, *et al.* Attractive In Situ Self-Reconstructed Hierarchical Gradient Structure of Metallic Glass for High Efficiency and Remarkable Stability in Catalytic Performance. *Advanced Functional Materials* **29**, (2019).
2. Hodge AM, Nieh TG. Evaluating abrasive wear of amorphous alloys using nanoscratch technique. *Intermetallics* **12**, 741-748 (2004).
3. Huang Y, Chiu YL, Shen J, Sun Y, Chen JJJ. Mechanical performance of metallic glasses during nanoscratch tests. *Intermetallics* **18**, 1056-1061 (2010).
4. Nie J, Wang Z, Ren Z, Li S, Chen X, Lin Wang Z. Power generation from the interaction of a liquid droplet and a liquid membrane. *Nat Commun* **10**, 2264 (2019).
5. Xu W, *et al.* A droplet-based electricity generator with high instantaneous power density. *Nature* **578**, 392-396 (2020).
6. Wu H, Wang Z, Zi Y. Multi-Mode Water-Tube-Based Triboelectric Nanogenerator Designed for Low-Frequency Energy Harvesting with Ultrahigh Volumetric Charge Density. *Advanced Energy Materials*, (2021).
7. Liu D, *et al.* Performance enhanced triboelectric nanogenerator by taking advantage of water in humid environments. *Nano Energy* **88**, (2021).
8. Zberg B, Uggowitzer PJ, Löffler JF. MgZnCa glasses without clinically observable hydrogen evolution for biodegradable implants. *Nat Mater* **8**, 887-891 (2009).
9. Sarac B, *et al.* Electrosorption of Hydrogen in Pd-Based Metallic Glass Nanofilms. *ACS Applied Energy Materials* **1**, 2630-2646 (2018).
10. Hu YC, *et al.* A Highly Efficient and Self-Stabilizing Metallic-Glass Catalyst for Electrochemical Hydrogen Generation. *Adv Mater* **28**, 10293-10297 (2016).
11. Jia Z, *et al.* Disordered Atomic Packing Structure of Metallic Glass: Toward Ultrafast Hydroxyl Radicals Production Rate and Strong Electron Transfer Ability in Catalytic Performance. *Advanced Functional Materials* **27**, (2017).
12. Wang B, *et al.* New Hydrophobic Organic Coating Based Triboelectric Nanogenerator for Efficient and Stable Hydropower Harvesting. *ACS Applied Materials & Interfaces* **12**, 31351-31359 (2020).
13. Ye C, *et al.* A Hydrophobic Self-Repairing Power Textile for Effective Water Droplet Energy Harvesting. *ACS Nano* **15**, 18172-18181 (2021).
14. Chatterjee S, *et al.* Enhanced sensing performance of triboelectric nanosensors by solid-liquid contact electrification. *Nano Energy* **77**, 105093 (2020).
15. Tang W, Chen BD, Wang ZL. Recent Progress in Power Generation from Water/Liquid Droplet Interaction with Solid Surfaces. *Advanced Functional Materials* **29**, (2019).
16. Sun YF, Wei BC, Wang YR, Li WH, Cheung TL, Shek CH. Plasticity-improved Zr-Cu-Al bulk

- metallic glass matrix composites containing martensite phase. *Applied Physics Letters* **87**, (2005).
17. Ding J, Inoue A, Zhu SL, Wu SL, Shalaan E, Al-Ghamdi AA. Formation, microstructure and mechanical properties of ductile Zr-rich Zr–Cu–Al bulk metallic glass composites. *Journal of Materials Research and Technology* **15**, 5452-5465 (2021).
 18. Niu S, Wang ZL. Theoretical systems of triboelectric nanogenerators. *Nano Energy* **14**, 161-192 (2015).
 19. Mi H-Y, *et al.* High-performance flexible triboelectric nanogenerator based on porous aerogels and electrospun nanofibers for energy harvesting and sensitive self-powered sensing. *Nano Energy* **48**, 327-336 (2018).
 20. Lin Z, *et al.* Rationally designed rotation triboelectric nanogenerators with much extended lifetime and durability. *Nano Energy* **68**, 104378 (2020).
 21. Chen P, *et al.* Super-Durable, Low-Wear, and High-Performance Fur-Brush Triboelectric Nanogenerator for Wind and Water Energy Harvesting for Smart Agriculture. *Advanced Energy Materials*, (2021).
 22. Wu H, Wang S, Wang Z, Zi Y. Achieving ultrahigh instantaneous power density of 10 MW/m² by leveraging the opposite-charge-enhanced transistor-like triboelectric nanogenerator (OCT-TENG). *Nat Commun* **12**, 5470 (2021).

REVIEWERS' COMMENTS

Reviewer #1 (Remarks to the Author):

The authors have correctly addressed all my concerns.

Reviewer #2 (Remarks to the Author):

Authors revised manuscript based on my comments and added good enough additional data to verify the stability of metallic glass. Now I believe the manuscript is ready for publication in Nature Communication. It is minor comment but it would be beneficial to add the micro-tensile test of MG to understand elastic strain range of current materials.

Reviewer #3 (Remarks to the Author):

The authors have addressed comments and revised the manuscript accordingly. It is acceptable for publication.

Reviewer #4 (Remarks to the Author):

The authors have addressed all the questions in detail.

Reviewer #1 (Remarks to the Author):

The authors have correctly addressed all my concerns.

Answer:

We sincerely thank the reviewer for giving the comments for our improvement and understanding the contributions of this work.

Reviewer #2 (Remarks to the Author):

Authors revised manuscript based on my comments and added good enough additional data to verify the stability of metallic glass. Now I believe the manuscript is ready for publication in Nature Communication. It is minor comment but it would be beneficial to add the micro-tensile test of MG to understand elastic strain range of current materials.

Answer:

Thank the reviewer for the professional suggestions. We added the investigation for the elastic strain range through the nano-indentation test.

The relationship between reduced modulus and elastic reduced modulus can be investigated through the following equation:²

$$\frac{1}{E_r} = \frac{1 - \nu^2}{E} + \frac{1 - \nu_i^2}{E_i} \quad (S.1)$$

Here, E_r is reduced modulus; E is elastic modulus; E_i is elastic modulus of diamond; ν_i is poison's ratio of diamond; ν is the poison's ratio, which can be assumed as 1/3.

After obtaining the elastic modulus, we simply assumed the yield strength as 1/3 of the hardness of materials. Then, the elastic strain limit ε can be calculated through the following equation:

$$\varepsilon = \frac{\text{yiled strength}}{\text{elastic modulus}} \quad (S.2)$$

The related parameters were obtained through the nano-indentation measurement, as depicted in Methods, and the final results of the elastic strain limit were summarized in the table below:

Table R1. Elastic strain limit of different MG samples.

Sample	E_r	ν	ν_i	E_i	E	Hardness	Yield	ε
--------	-------	-------	---------	-------	-----	----------	-------	---------------

	(Gpa)			(GPa)	(GPa)	(GPa)	strength (GPa)	
Zr ₄₅ Cu ₄₀ Al ₁₅	107.16	1/3	0.07	1100	105.48	7.81	2.604	0.0247
Zr ₅₀ Cu ₄₀ Al ₁₀	91.99	1/3	0.07	1100	89.20	6.56	2.186	0.0246
Zr ₄₅ Cu ₃₅ Al ₂₀	110.33	1/3	0.07	1100	108.94	7.69	2.562	0.0236

It can be noticed from Supplementary R6 that the elastic strain limit of the three MG samples was around 2.36%~2.47%, which was much larger than the involved elastic strain in our experiment calculated previously which is in ~‰ scale, further confirming the stability of the MG samples in our experiments.

Reviewer #3 (Remarks to the Author):

The authors have addressed comments and revised the manuscript accordingly. It is acceptable for publication.

Answer:

We sincerely thank the reviewer for the professional suggestions for improving our manuscript.

Reviewer #4 (Remarks to the Author):

The authors have addressed all the questions in detail.

Answer:

We sincerely appreciate the reviewer for understanding the novelty, significance and contributions of this work.

References for response only:

1. Bao YW, Wang W, Zhou YC. Investigation of the relationship between elastic modulus and hardness based on depth-sensing indentation measurements. *Acta Materialia* **52**, 5397-5404 (2004).